# Short-Term Resilience of Soil Microbial Communities and Functions Following Severe Environmental Changes

Stefano Mocali [1,*], Antonio Gelsomino [2], Paolo Nannipieri [3], Roberta Pastorelli [1], Laura Giagnoni [4], Beatrix Petrovicova [2] and Giancarlo Renella [5]

1 Research Centre for Agriculture and Environment, Council for Agricultural Research and Economics, Via di Lanciola 12/A, Cascine del Riccio, 50125 Firenze, Italy; roberta.pastorelli@crea.gov.it

2 Department of Agricultural Sciences, Mediterranean University of Reggio Calabria, Feo di Vito, 89122 Reggio Calabria, Italy; agelsomino@unirc.it (A.G.); beatrix.petrovicova@unirc.it (B.P.)

3 Department of Agriculture, Food, Environment and Forestry, University of Florence, P.le delle Cascine, 28, 50144 Firenze, Italy; paolo.nannipieri@unifi.it

4 Department of Civil Engineering, Architecture, Environmental and Mathematics (DICATAM), University of Brescia, Via Branze, 43, 25123 Brescia, Italy; laura.giagnoni@unibs.it

5 Department of Agronomy, Food, Natural Resources Animals and Environment (DAFNAE), University of Padua, Viale dell'Università 16, Legnaro, 35020 Padova, Italy; giancarlo.renella@unipd.it

* Correspondence: stefano.mocali@crea.gov.it

**Abstract:** Soil microorganisms are key drivers of soil biochemical processes, but the resilience of microbial communities and their metabolic activity after an extreme environmental change is still largely unknown. We studied structural (bacterial and fungal communities) and functional responses (soil respiration, adenosine triphosphate (ATP) content, hydrolase activities involved in the mineralization of organic C, N, P and S, and microbial community-level physiological profiles (CLPPs)) during the microbial recolonization of three heat-sterilized forest soils followed by cross- or self-reinoculation and incubation for 1, 7 and 30 days. Soil ATP content, biochemical activities and CLPP were annihilated by autoclaving, whereas most of the hydrolase activities were reduced to varying extents depending on the soil and enzyme activity considered. During the incubation period, the combination of self- and cross-reinoculation of different sterilized soils produced rapid dynamic changes in enzymatic activity as well as in microbial structure and catabolic activity. Physicochemical properties of the original soils exerted a major influence in shaping soil functional diversity, while reinoculation of sterilized soils promoted faster and greater changes in bacterial community structure than in fungal communities, varying with incubation period and soil type. Our results also confirmed the importance of microbial richness in determining soil resilience under severe disturbances. In particular, the new microbial communities detected in the treated soils revealed the occurrence of taxa which were not detected in the original soils. This result confirmed that rare microbial taxa rather than the dominant ones may be the major drivers of soil functionality and resilience.

**Keywords:** microbial diversity; enzymes; catabolic activity; soil recolonization; sterilization; resilience

## 1. Introduction

Soil microbial communities display high metabolic diversity and functional redundancy, two features that make them major drivers of nutrient biogeochemical cycles and globally a key factor of soil resilience [1–4]. Despite the fast proliferation rates and high colonization capacities of microbial species, the long-held historical view that "*everything is everywhere, but the environment selects*" [5] is no longer accepted in soil microbial ecology to explain the complex interactions occurring among environmental characteristics, microbial community structure and microbial functional activity. In fact, large-scale studies of soil microbial distribution supported by 'omics' methods have led to the development of new conceptual frameworks of species assembly, such as the 'coalescence' [6] and 'metacommunity' theories [7]. The coalescence theory proposes that previously separated microbial

communities can completely mix in the soil environment and form a new community, whereas the metacommunity theory describes the diversity of microbial communities in terms of species compatibility with soil physicochemical properties and microbial interactions of individuals or species by stochastic and deterministic dispersal processes.

In actual agricultural contexts, where soils are often threatened by anthropic and environmental factors, it is extremely important to better understand the role of soil microbiota in maintaining soil functions, even after severe disturbances. In fact, although soil-borne microorganisms possess high metabolic flexibility and display tolerance to changing environmental conditions [8], the response of soil microbial communities to external disturbance or environmental changes is still poorly understood [9–12]. Experiments with soil sterilization and soil mixing have been considered a suitable approach to understand the recovery of microbial diversity and soil functions after extreme impacts and variations induced by the coalescence of soil communities. Previous studies on soil microbial recolonization after sterilization by fumigation, autoclaving or γ-irradiation [13–16] have shown that microbial metabolic activities are primarily involved in the colonization capacity of microbial communities of the same and/or different sterilized soils. Complex ecological interactions as well as adaptive responses between microbial species during soil recolonization have been also reported [17,18]. However, few studies have assessed the recovery of bacterial and fungal community structure and functions during the recolonization of sterilized soils. For example, Latour et al. [19] reported that the composition of a mixed *Pseudomonas* community developed differently when inoculated into sterile soils with different characteristics. Delmont et al. [20] observed that distinct microbial communities from soils of different locations and land use evolved similarly when recolonizing the same sterilized soil. The combination of self- and cross-inoculation of different soils and sterilized soils showed how the new microbial community structure is shaped by soil properties and levels of fertilization, confirming the importance of nutrient availability as a key factor shaping the composition of mixed microbial communities [21,22]. In a microcosm experiment, Wertz et al. [18] modified the soil microbial community composition by serial dilution and reinoculation of sterile soil. However, they observed no effects of microbial community composition on soil functionality. In a soil cross-inoculation experiment, Kapagianni et al. [23] showed that the soil type had a major influence on the composition of the new soil microbial communities, whereas enzymatic activities were related to the inoculum source initially and to the soil pH value at later stages.

Based on such studies, two central questions still have no univocal answer: (i) do soil microbial communities recover to their original structure and functionality after drastic disturbance? and (ii) do the same microbial communities colonizing different soils express similar biochemical functions? We hypothesized that microbial communities originating from different soils are able to recolonize the same soil after a drastic disturbance or even colonize soils with different physicochemical properties and that the newly introduced microbial communities are capable of expressing their metabolic potential in different soil types. We tested these hypotheses in a laboratory incubation experiment using a combination of sterilization and self- and cross-inoculation of three soils with contrasting pH, texture and organic C contents.

We measured the microbial biomass, soil respiration and N mineralization and the activity of soil enzymes involved in C, N, P and S mineralization and related them to the diversity of bacterial and fungal community structure and the community-level physiological profile (CLPP) of the culturable fraction of soil microbial communities in all sterilized non-inoculated, self- and cross-inoculated soils.

The present work can improve knowledge of the resilience of soil microbial community structure and functions, including potential $CO_2$ emission and N mineralization, catabolic activity and enzymatic activity in newly colonized soils. Improvement of base knowledge on soil microbial ecology can also be important in the current scenario of climate change, in which increased intensity and/or frequency of extreme drought and rain events may lead to more drastic alterations of soil microbial community structure with unknown consequences

on soil microbial diversity and metabolic activity [24], and to evaluate the potential of soil reclamation interventions.

## 2. Materials and Methods

### 2.1. Soils and Soil Treatments

Soils with contrasting physicochemical properties (Table 1) were collected from the A horizon of three forest sites. The Vallombrosa soil (Val) was collected from a protected silver fir (*Abies alba* Mill.) forest (43°43′58″ N, 11°33′23″ E, 950 m a.s.l.) which developed on Oligocene sandstone and is classified as fine-loamy, mixed, mesic Fragic Dystrudept (Soil Survey Staff, 2010). The Romola soil (Rom) was collected from former arable land (43°41′53″ N, 11°09′41″ E, 205 m a.s.l.) abandoned for 40 years, vegetated with mixed shrubs and herbs and dominated by holm oak (*Quercus ilex* L.), formed on alluvial deposits, and is classified as coarse, mixed, thermic Eutric Cambisol (Soil Survey Staff, 2010). The Vicarello soil (Vic) was sampled from the CREA experimental station in Volterra (43°36′48″ N, 10°27′53″ E, 150 m a.s.l.), developed on Pliocene clayey marine deposits, is classified as fine, mixed, thermic Vertic Xerochrept (Soil Survey Staff, 2010), and was sampled from 50-year-old mixed woodland vegetation dominated by a downy oak (*Quercus pubescens* Wild.).

**Table 1.** Main physical and chemical properties of Vallombrosa (Val; acidic loamy forest), Romola (Rom; sandy arable) and Vicarello (Vic; clay calcareous forest) soils. Values are means $\pm$ SD (n = 3).

| Soil | Sand | Silt | Clay | pH | TOC | TN | CEC | TCa | ACa | NH$_4^+$-N | NO$_3^-$-N | TOP | Olsen-P |
|------|------|------|------|-----|-----|-----|-----|-----|-----|-----------|-----------|-----|---------|
| | % | | | | g kg$^{-1}$ | | cmol$_c$ kg$^{-1}$ | g kg$^{-1}$ | | mg kg$^{-1}$ | | | |
| Val | 48.9 | 33.0 | 18.1 | 5.0 ± 0.2 | 36.6 ± 1.5 | 2.2 ± 0.3 | 26.6 ± 0.8 | 0 | 0 | 29.8 ± 3.6 | 25.7 ± 5.1 | 34.1 ± 2.8 | 7.4 ± 2.9 |
| Rom | 90.7 | 3.6 | 5.7 | 6.7 ± 0.1 | 10.5 ± 0.3 | 0.98 ± 0.2 | 16.9 ± 0.5 | 0 | 0 | 13.0 ± 2.5 | 22.3 ± 1.4 | 20.9 ± 1.9 | 8.7 ± 3.3 |
| Vic | 20.5 | 37.3 | 42.2 | 8.0 ± 0.1 | 22.9 ± 0.3 | 2.2 ± 0.2 | 25.3 ± 0.7 | 128 ± 5 | 83 ± 3 | 21.7 ± 1.6 | 14.7 ± 2.0 | 22.0 ± 1.8 | 6.2 ± 0.3 |

Soil variables are: pH; TOC, total organic C; TN, total N; CEC, cation-exchange capacity; TCa, total soil carbonate; ACa, active soil carbonate; NH$_4^+$-N, ammonium-N; NO$_3^-$-N, nitrate-N; TOP, total organic P; Olsen-P, Olsen-extractable P.

After sampling, field moist soils were sieved at <2 mm particle size, brought to 50% water-holding capacity (WHC) and then conditioned at 25 °C in the dark for 7 days to stabilize the microbial activity. After conditioning, an amount of soil equivalent to 1 kg dry weight of each soil was autoclaved (121 °C, 1 bar, 1 h) in glass Petri dishes containing 100 g of soil each, incubated in the dark (25 °C, 24 h) and then further autoclaved (121 °C, 1 bar, 1 h) according to Wolf and Skipper [25].

We sterilized the original soils by autoclaving because soil sterilization with γ-irradiation or fumigants (e.g., chloroform) leaves most of the soil enzymes still active [15,26–29], whilst autoclaving reduces most soil enzyme activities below detection levels, allowing us to assess the newly microbial-derived enzyme production during the recolonization of sterilized soils [30]. Each autoclaved soil ('recipient soil') was reinoculated in a factorial way with the same (self-reinoculation) or the other (cross-inoculation) non-sterile fresh soil at 5% (*w/w*) rate and then thoroughly mixed until a homogeneous incorporation had been reached. Control treatments were both autoclaved non-inoculated soils (Val*, Vic*, Rom*) and non-autoclaved non-inoculated fresh soils (Val, Vic, Rom). After sterilization and inoculation, 100 g of each soil were placed into 1 L air-tight glass flasks provided with 3-way valves for head-space gas sampling and incubated in the dark at 25 °C for 24 h (day 1), 7 and 30 days. Based on the results provided by similar experiments [20–23], we choose an incubation time of 30 days to focus on the very first changes occurring after the disturbance. All treatments were prepared as three independent replicates for each treatment and incubation time, and at each sampling time the incubated samples were destructively sampled and immediately used for chemical and biochemical analyses, CLPP and microbial community fingerprinting.

## 2.2. Soil Respiration, N Ammonification, Microbial Biomass and Enzymatic Activities

Soil respiration was determined by the quantification of CO2 emission by head-space gas sampling and gas chromatographic analysis (Hewlett-Packard Model 6890, equipped with a thermal conductivity detector and a packed column (Porapak Q, Supelco, Bellefonte, PA, USA)), according to Blackmer and Bremner [31]. Three control jars with no soil were used to correct for atmospheric $CO_2$-C background concentration. N ammonification was determined by extractions with 1 M KCl (1:5, w:v), followed by colorimetric determination of $NH_4^+$-N concentration with the Nessler reagent. Soil microbial biomass was estimated by the adenosine 5′-triphosphate (ATP) soil content, determined according to Ciardi and Nannipieri [32]. Acid and alkaline phosphomonoesterase, β-glucosidase, protease and urease were determined with colorimetric assays, as described by Dick et al. [33].

## 2.3. Community-Level Physiological Profile (CLPP) Fingerprinting

The community-level physiological profile (CLPP) was analyzed using the Biolog Ecoplates® for bacteria and the FF Microplates® for fungi (Biolog Inc., Hayward, California, USA). Each plate was inoculated with 150 μL per well of a 10−1 (w/v) soil suspension obtained in sterile NaCl solution (9 g $L^{-1}$) after bead beating at 250 rpm for 30 min. Then, soil suspensions were fortified with cycloheximide (100 μg $mL^{-1}$) or kanamycin (100 μg $mL^{-1}$) to selectively favour bacterial (Biolog Ecoplates®) or fungal (FF Microplates®) growth, respectively. Plates were incubated at 30 °C in the dark for 190 h. The metabolic activity of the microbial communities was monitored by recording the absorbance at 590 nm (OD590) at 12 h intervals using a Biolog Microstation System and expressed as average well color development (AWCD) [34]. The area of each curve was calculated according to Guckert et al. [35], whereas the catabolic versatility (CV) index and the kinetic 's' parameter (the time to the midpoint of the exponential portion of the curve) were calculated according to Burkhardt et al. [36] and Lindstrom et al. [37], respectively.

## 2.4. Soil DNA Extraction and PCR-DGGE Fingerprinting

Total soil DNA was extracted from 250 mg moist soil using the Fast DNA™ SPIN Kit for soil (MP Biomedicals™, Santa Ana, CA, USA). Yield and quality of soil-extracted DNA were checked by 0.7% agarose gel electrophoresis and quantified according to Marstorp and Witter [38]. For DGGE analysis of the bacterial community, the PCR amplification of soil-extracted DNA was performed using the primer system flanking the hypervariable V6–V8 region of 16S rDNA targets, F968-GC/R1401 [39]. For DGGE analysis of the fungal community, the DNA was amplified by a nested-PCR using specific primers for 18S rDNA targets, NS1f/FR1r for the first step and FF390f/FR1r-GC for the second step [40]. Amplification reactions were carried out using an iCycler™ thermal cycler (Bio-Rad Laboratories, Hercules, California, USA) under the following conditions: 94 °C for 4 min, followed by 35 cycles consisting of denaturing at 95 °C for 45 s, annealing at 55 °C (bacteria) or at 48° C (fungi) for 45 s, extension at 72° C for 45 s and a final extension step at 72 °C for 7 min. The DGGE analysis was carried out using an INGENY phorU-2 System (Ingeny International BV Goes, NL, USA). An amount of 500 ng of amplified DNA was loaded onto a 6% (w/v) polyacrylamide (acrylamide/bis-acrylamide 37.5:1) gel, containing a linear chemical gradient ranging from 50–65% denaturant for 16S rDNA (bacterial) amplicons or from 46–58% denaturant for 18S rDNA (fungal) amplicons (100% denaturant corresponds to 40% (v:v) deionized formamide plus 7 M urea). The electrophoresis was run in 1 × TAE buffer at 60 °C at a constant voltage of 80 V for 17 h. After the run, the gels were stained with SYBR® Gold Nucleic Acid Gel Stain (Thermo Fisher Scientific, Waltham, MA, USA). Gel images were digitally captured under UV light (λ = 302 nm) using the ChemiDoc Imaging System (Bio-Rad Laboratories).

## 2.5. Data Analysis

Biochemical data are mean values from three independent soil replicates (n = 3) and are expressed on an oven-dry weight basis (105 °C, 24 h). After testing for deviation

from normality (Kolmogorov–Smirnov test) and homogeneity of within-group variances (Levene's test), data were analyzed using standard analysis of variance (ANOVA) (SPSS v. 11.0, IBM, Armonk, New York, NY, USA). Bacterial and fungal CLPP data were analyzed by NMDS (Manhattan index) by means of PAST2 software [41], and the accuracy of the plots was determined by calculating a 2D stress value. In order to simplify the analysis, all the CLPP data were divided into different categories, according with Insam [42], and analyzed by two-way-ANOVA (treatment and time). DGGE banding patterns were analyzed by GelCompar® II software v 4.6 (Applied Maths, Sint-Martens-Latem, Belgium). Normalization of bands within and between gels was conducted by defining an active reference system. The number of bands (species richness) and their relative abundance (Shannon–Weiner and Simpson indices) within each DGGE profile were used as a proxy of richness and diversity of soil bacterial and fungal communities and calculated as described by Pastorelli et al. [43]. Bands with a minimum area below 1% were excluded from the calculations. For each DGGE, a binary matrix based on the position and presence/absence of bands in the different profiles was generated and imported into PAST2 software [41] for multivariate statistical analysis. NMDS, based on the Dice similarity coefficient, was performed to represent the similarity distance between each DGGE profile in a two-dimensional space. One-way analysis of similarity (ANOSIM) and permutational multivariate analysis of variance (PERMANOVA), based on Dice similarity coefficient and 9999 permutational tests, were performed in order to test whether differences in the assemblage grouping observed in NMDS plots were significant. Pairwise Pearson correlation and principal component analysis (PCA) were used to statistically process the entire dataset, including soil chemical, biochemical, molecular and CLPP data.

## 3. Results

### 3.1. Soil Respiration, N Ammonification, ATP and Microbial Biomass

Heat-sterilization drastically reduced soil respiration; however, it recovered after reinoculation and incubation with trends similar in all treatments but different in values depending on the recipient soil and the inoculum source (Figure 1A). At the end of the incubation period, the Rom soil showed the lowest cumulative respiration values compared to Vic and Val soils.

The highest N ammonification values were observed in Val soils, whether reinoculated or not, compared to the control Val soil, with values which were up to three-fold those of the Rom and Vic reinoculated soils. Control soils, whether sterilized or not, showed similar ammonification values across the entire incubation period (Figure 1B). Autoclaving immediately reduced the soil ATP to undetectable concentrations in all sterilized soils (Val*, Rom*, Vic*), and values three- or two-fold lower than in the respective pristine soils were observed in Vic and Rom soils, respectively, after 7 days of incubation (Figure 1C). In contrast, in self- and cross-inoculated Val soils the ATP content remained significantly lower than the respective non-autoclaved and non-reinoculated soil over time. At the end of the incubation period the ATP content in all Val* and self- and cross-inoculated Val* soils remained significantly lower than Val soil, whereas Rom* and Vic* soils reached final values similar to those of the respective intact soils. Rom* and Vic* cross-inoculated soils remained significantly lower, whereas cross-inoculation increased the ATP concentrations to levels depending on the recipient soil type and inoculum source (Figure 1C).

The microbial biomass (dsDNA) values were almost zero in all the sterilized soils at day 1. Then, whereas Rom and Val soils showed increasing dsDNA values over time, Vic sterilized and reinoculated soils showed the highest values (up to 40–50 mg/kg in Vic*+Vic and Vic*+Val), just after 7 days, which slightly decreased at the end of incubation.

Interestingly, all the Val* soils (reinoculated or not), displayed lower values than the untreated control. In contrast, after 7 and 30 days the untreated soils of Rom and Vic showed significantly lower values than the sterilized and reinoculated soils (Figure S1).

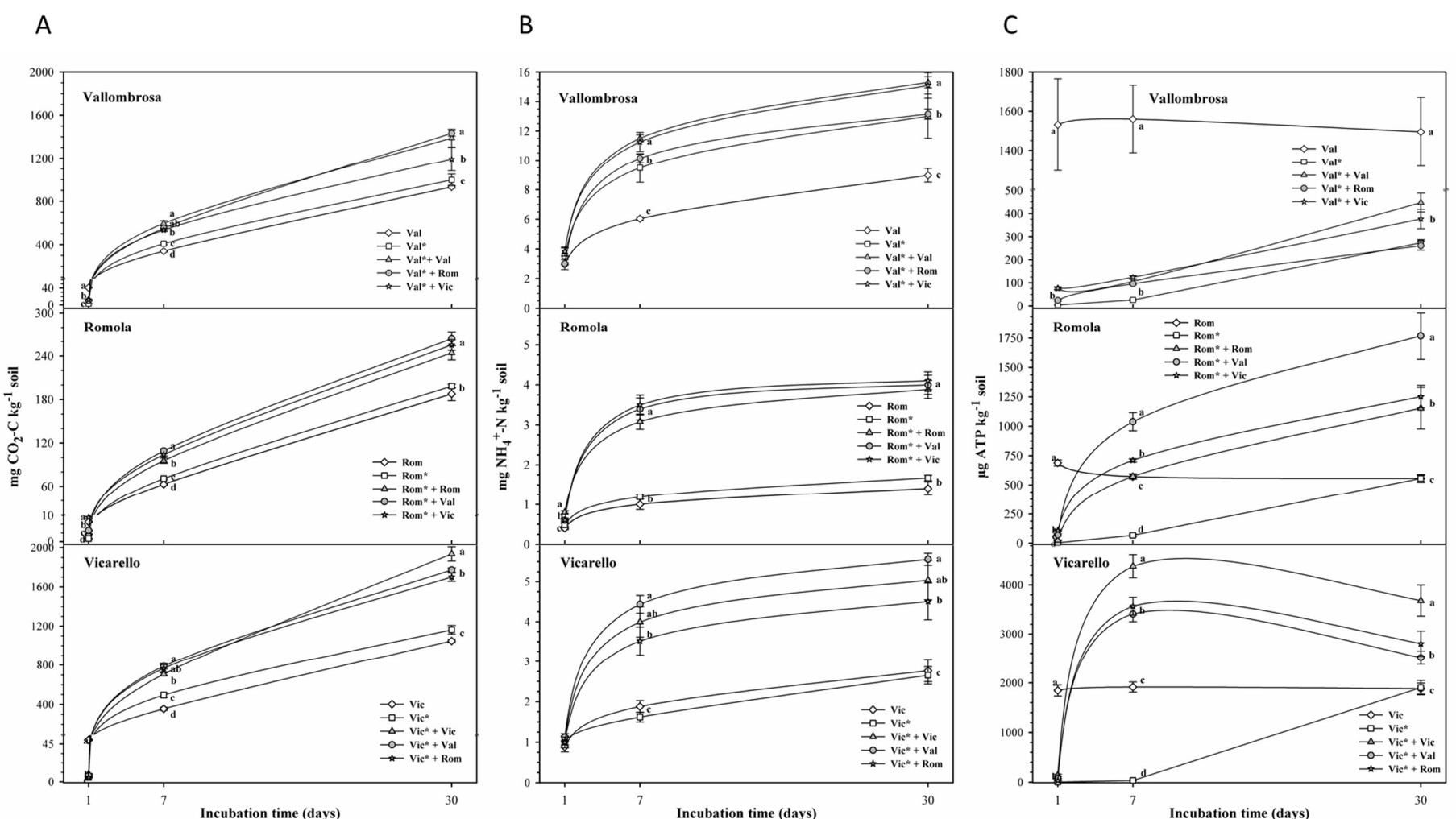

**Figure 1.** (**A**) Cumulative soil respiration in reinoculated soils during a 30 d incubation experiment. (**B**) Ammonification in self- or cross-reinoculated soils during a 30 d incubation experiment. At each sampling time, different letters indicate a significant ($p < 0.05$) difference among means (n = 3). (**C**) ATP content in reinoculated soils during a 30 d incubation experiment. Heat-sterilized and non-inoculated soils (Val*, Rom*, Vic*) together with non-heat-sterilized and non-reinoculated soils (Val, Rom, Vic) were taken as control treatments. At each sampling time, different letters indicate a significant ($p < 0.05$) difference among means (n = 3).

*3.2. Soil Enzymatic Activities*

Soil hydrolytic activities showed different values and trends depending on the soil type, soil reinoculation and incubation times.

### 3.2.1. Enzymatic Activities in the Vallombrosa Soil

After sterilization, the enzymatic activities of Val soils were at undetectable levels, whereas all activities except β-glucosidase recovered immediately following reinoculation (Table 2). After 7 days of incubation, the alkaline phosphomonoesterase was still undetectable in the Val* soil, at significantly lower values in the Val*+Val and Val*+Rom soils, and at a higher level in the Val*+Vic (+167%) as compared to the untreated control soil. After 30 days, the alkaline phosphomonoesterase activity was significantly higher in all the sterilized and inoculated soils compared to the untreated control soil. After 7 days of incubation, the acid phosphomonoesterase activity displayed lower values in the Val* soil and significantly higher in the Val*+Val, Val*+Rom and Val*+Vic soils as compared to the control soil. On the other hand, it reached the same level of the control soil in all sterilized and reinoculated soils after 30 days of incubation.

The β-glucosidase activity showed higher levels in all sterilized and reinoculated soils after both 7 and 30 days of incubation as compared to the control soil (Table 2), with the highest values detected in Val*+Rom soil after 30 days. The protease and urease activities showed significantly lower values in all sterilized and inoculated Val* soils as compared to the control soil after both 7 and 30 days of incubation (Table 2).

### 3.2.2. Enzymatic Activities in the Romola Soil

Sterilized Rom* samples showed undetectable levels of hydrolytic activity after 1 day of incubation, whereas variable levels of enzymatic activity were detected in all Rom* reinoculated soils (Table 2). After 7 days of incubation, the alkaline phosphomonoesterase in the Rom* and Rom*+Val soils was significantly lower than untreated Rom. The Rom*+Vic soil showed significant higher values as compared to the control soil, whereas the Rom*+Rom soil showed similar results than the untreated Rom soil. After 30 days of incubation, the alkaline phosphomonoesterase activity was significantly higher in all the inoculated soils as compared to the control soils. The acid phosphomonoesterase activity was undetectable in Rom* soil after 7 days of incubation, at significantly lower values in the Rom*+Rom soil and at the same level of the control soil in the Rom*+Val and Rom*+Vic soils. After 30 days of incubation, the acid phosphomonoesterase activity in the Rom* and reinoculated Rom soils was lower than that detected in the Rom soil. The β-glucosidase activity was at the same level as in the control soils and in Rom*+Vic soil after both 7 and 30 days of incubation, whereas it was lower in the Rom*+Rom and Rom*+Val soils as compared to the control soils. The protease and urease activities were at significantly lower levels in all sterilized and reinoculated Rom* soils as compared to the control soil after both 7 and 30 days of incubation.

**Table 2.** Enzymatic activities in self- or cross-reinoculated heat-sterilized soils during a 30 d incubation experiment. Heat-sterilized and non-inoculated soils (Val*, Rom*, Vic*) together with non-heat-sterilized and non-reinoculated soils (Val, Rom, Vic) were taken as control treatments. At each sampling time, different letters indicate a significant ($p < 0.05$) difference among means (n = 3).

| Enzymatic Activity | Time (days) | Vallombrosa | | | | | Romola | | | | | Vicarello | | | | |
|---|---|---|---|---|---|---|---|---|---|---|---|---|---|---|---|---|
| | | Val | Val* | Val*+ Val | Val*+ Rom | Val*+ Vic | Rom | Rom* | Rom*+ Rom | Rom*+ Val | Rom*+ Vic | Vic | Vic* | Vic*+ Vic | Vic*+ Val | Vic*+ Rom |
| Alkaline phosphomo-noesterase (mg $p$-NP [a] kg$^{-1}$ h$^{-1}$) | 1 | 3849 a | BDL | 668 b | BDL | 541 b | 7031 a | BDL | 89 b | 1110 c | 334 d | 17,501 a | BDL | 2323 b | BDL | BDL |
| | 7 | 3963 a | BDL | 1845 b | 1570 b | 10,589 c | 6704 a | 2113 b | 7248 a | 2921 b | 15,169 c | 18,019 a | BDL | 17,199 a | 14,625 a | 17,082 a |
| | 30 | 3977 a | 9228 b | 8790 b | 6803 c | 11,597 d | 6304 a | 6645 a | 7624 a | 8811 a | 11,059 b | 17,452 a | 10,050 b | 11,201 b | 14,945 c | 6127 d |
| Acid phosphomo-noesterase (mg $p$-NP [a] kg$^{-1}$ h$^{-1}$) | 1 | 22,732 a | BDL | 2093 b | BDL | BDL | 2092 a | BDL | 90 b | 763 c | 700 c | 5396 a | BDL | 1016 b | 1848 c | BDL |
| | 7 | 22,274 a | 9282 b | 37,036 c | 42,979 c | 35,455 c | 2351 a | BDL | 678 b | 2396 a | 2251 a | 5277 a | BDL | 2174 b | 2260 b | 2446 b |
| | 30 | 21,833 a | 21,930 a | 25,507 b | 23,080 ab | 18,315 c | 2238 a | 774 b | 844 b | 1774 a | 1720 a | 5287 ab | 3314 c | 4915 a | 6494 b | 2979 c |
| β-Glucosidase (mg $p$-NP [a] kg$^{-1}$ h$^{-1}$) | 1 | 2492 a | BDL | BDL | BDL | BDL | 1641 a | BDL | 15 b | 247 c | 1004 d | 5853 a | BDL | 1116 b | BDL | BDL |
| | 7 | 2588 a | 4054 a | 19,568 b | 17,899 b | 8133 c | 1583 a | 1574 a | 1330 a | 991 b | 1768 a | 6448 a | 2805 b | 4246 c | 4136 c | 4123 c |
| | 30 | 2845 a | 17,584 b | 18,044 b | 23,453 c | 11,293 d | 1531 a | 1583 a | 1090 b | 878 b | 1697 a | 4946 a | 776 b | 960 b | 441 b | 355 c |
| Protease (mg NH$_4^+$-N kg$^{-1}$ h$^{-1}$) | 1 | 124.2 a | BDL | 19.0 b | 30.1 c | 14.6 b | 117.0 a | BDL | 19.6 b | 29.5 c | 10.6 d | 106.4 a | BDL | BDL | BDL | BDL |
| | 7 | 101.5 a | BDL | 16.9 b | 17.7 b | 25.5 c | 121.8 a | BDL | 23.6 b | 21.0 b | 26.6 b | 100.9 a | BDL | BDL | BDL | BDL |
| | 30 | 108.8 a | 26.5 b | 16.3 c | 12.2 c | 17.0 d | 110.7 a | 3.7 b | 18.9 c | 26.0 d | 10.2 e | 91.1 a | 22.5 b | 44.1 c | 28.4 b | 40.5 c |
| Urease (mg NH$_4^+$-N kg$^{-1}$ h$^{-1}$) | 1 | 82.9 a | BDL | 11.7 b | 7.7 b | 10.8 b | 101.0 a | BDL | 14.7 b | 3.6 c | 12.9 b | 7.4 a | BDL | BDL | BDL | BDL |
| | 7 | 78.4 a | 6.3 b | 22.8 c | 5.9 b | 34.8 d | 96.0 a | BDL | 14.7 b | 2.0 c | 12.8 b | 7.4 a | BDL | 13.3 a | 124.4 b | 64.7 c |
| | 30 | 69.4 a | 14.7 b | 24.3 c | 19.7 bc | 38.9 d | 87.9 a | 3.2 b | 13.9 c | 4.3 b | 14.7 c | 7.5 a | 43.2 b | 8.2 a | 137.8 c | 49.2 b |

[a] $p$-NP, $p$-nitrophenol; BDL, below detection limit.

### 3.2.3. Enzymatic Activities in the Vicarello Soil

No hydrolytic activities were found in Vic* and Vic*+Rom treatments after 1 day of incubation (Table 2), whereas detectable enzymatic activities were: the alkaline and acid phosphomonoesterase and β-glucosidase activities detected in the Vic*+Vic soil and acid phosphomonoesterase detected in the Vic*+Val soil. After 7 days of incubation, the alkaline phosphomonoesterase was still not detectable in the Vic* soil, whereas it was found in the Vic*+Vic, Vic*+Val* and Vic+Rom soils at similar levels as in Vic soil. After 30 days of incubation, the alkaline phosphomonoesterase was detected in all the samples. Remarkably, Vic*, Vic*+Vic and Vic*+Rom soils showed significantly lower values than in the control soil, whereas in the Vic*+Val* soils it was at similar levels as after 7 days. The acid phosphomonoesterase was detected at significantly lower values in all the sterilized and reinoculated soils after both 7 and 30 days of incubation, with the exception of the Vic*+Val, which showed significantly higher values as compared to the control soil (Table 2). The β-glucosidase activity was significantly lower in all the sterilized and inoculated soils after 7 and 30 days of incubation. The protease activity was not detectable after 7 days of incubation and was at significantly lower levels in all sterilized and inoculated Vic* soils as compared to Vic soil after both 7 and 30 days of incubation. Urease activity was not detectable after 7 days of incubation in the Vic* soil, whereas it was detected at higher levels in the inoculated soils. After 30 days of incubation, it was at higher levels in all sterilized soils as compared to the non-sterilized Vic soil, with the exception of the Vic*+Vic soil, which displayed the same value as the control soil.

### 3.3. Soil Community-Level Physiological Profile

The CLPP of the three soils showed different patterns and trends depending on the soil, self- or cross-reinoculation and the incubation time. CLPP data for bacteria and fungi were reported as AWCD (Figure 2) as well as grouped according to their substrate utilizations (Tables 3 and 4). Overall, the CLPP analysis showed that after 7 days of incubation, the sterilized and reinoculated soils had higher AWCD values than the respective control soils. In general, Val soils displayed the greater catabolic activity, and the highest bacterial AWCD values were obtained in Val*+Rom soils after 7 and 30 days.

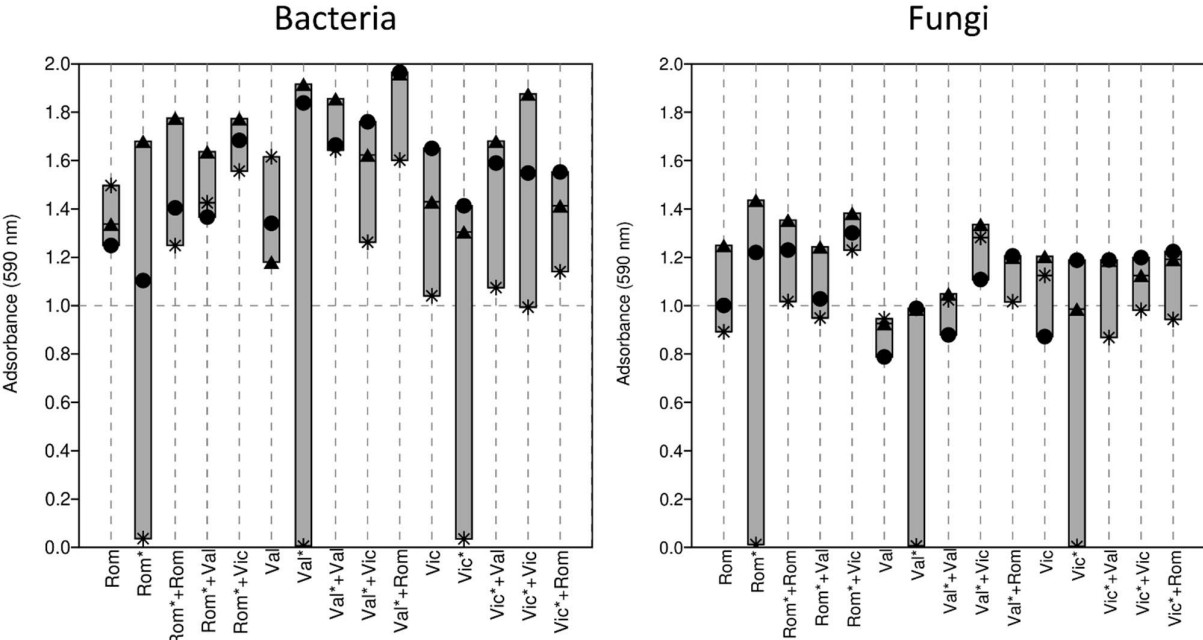

**Figure 2.** Biolog average well color development (AWCD) values for bacteria (EcoPlates) and fungi (FF microplates) from self- or cross-reinoculated heat-sterilized Romola, Vallombrosa and Vicarello soils after 1 day (star), 7 days (triangle) and 30 days of incubation (circle).

**Table 3.** Bacterial catabolic activity of self- or cross-inoculated soils during a 30 d incubation experiment. Heat-sterilized and non-inoculated soils (Val*, Rom*, Vic*) together with non-sterilized and non-reinoculated soils (Val, Rom, Vic) were taken as control treatments. At each sampling time, different letters indicate a significant ($p < 0.05$) difference among means (n = 3).

| Substrate Category | Time (days) | Soil Treatments | | | | | | | | | | | | | | |
| --- | --- | --- | --- | --- | --- | --- | --- | --- | --- | --- | --- | --- | --- | --- | --- | --- |
| | | Vallombrosa | | | | | Romola | | | | | Vicarello | | | | |
| | | Val | Val* | Val*+Val | Val*+Rom | Val*+Vic | Rom | Rom* | Rom*+Rom | Rom*+Val | Rom*+Vic | Vic | Vic* | Vic*+Vic | Vic*+Val | Vic*+Rom |
| Amines | 1 | 1.790 a | 0.000 b | 1.613 a | 1.720 a | 1.148 ab | 1.116 a | 0.018 b | 1.217 a | 1.167 a | 1.510 a | 0.926 a | 0.000 b | 0.942 a | 0.860 a | 0.902 a |
| | 7 | 1.215 a | 1.521 a | 2.028 b | 1.601 a | 1.585 a | 1.233 a | 1.675 b | 1.560 b | 1.492 b | 1.564 b | 1.438 a | 1.347 a | 1.662 a | 1.756 a | 1.317 a |
| | 30 | 1.600 a | 2.056 b | 1.757 a | 1.915 ab | 1.559 a | 1.277 a | 1.092 a | 1.376 a | 1.195 a | 1.648 b | 1.527 a | 1.626 a | 1.365 a | 1.789 a | 1.560 a |
| Amino acids | 1 | 2.030 a | 0.000 b | 1.835 a | 1.872 a | 1.499 a | 1.819 a | 0.055 b | 1.614 a | 1.716 a | 1.847 a | 1.362 a | 0.024 b | 1.168 a | 1.382 a | 1.421 a |
| | 7 | 1.297 a | 2.035 ab | 2.019 ab | 2.226 b | 1.815 ab | 1.601 a | 1.743 a | 1.931 a | 1.572 a | 1.909 a | 1.847 a | 1.546 a | 2.189 a | 1.850 a | 1.670 a |
| | 30 | 1.553 a | 1.963 ab | 2.146 b | 2.147 b | 2.006 ab | 1.616 a | 1.483 a | 1.530 a | 1.542 a | 1.684 a | 1.790 a | 1.638 a | 1.745 a | 1.751 a | 1.748 a |
| Carbohydrates | 1 | 1.630 a | 0.005 b | 1.566 a | 1.690 a | 1.266 a | 1.445 a | 0.044 b | 1.203 a | 1.324 a | 1.546 a | 1.016 a | 0.008 b | 0.883 a | 1.007 a | 1.071 a |
| | 7 | 1.519 a | 1.950 a | 1.827 a | 1.923 a | 1.396 a | 1.357 a | 1.619 a | 1.837 b | 1.727 ab | 1.675 a | 1.313 a | 1.325 a | 1.986 a | 1.644 a | 1.354 a |
| | 30 | 1.330 a | 1.830 b | 1.534 b | 1.801 b | 1.726 b | 1.346 a | 1.215 a | 1.542 a | 1.438 a | 1.671 b | 1.463 a | 1.559 a | 1.526 a | 1.641 a | 1.782 a |
| Carboxylic acids | 1 | 1.441 a | 0.000 b | 1.512 a | 1.513 a | 1.079 a | 1.469 a | 0.037 b | 1.146 a | 1.308 a | 1.413 a | 0.757 a | 0.018 b | 0.748 a | 0.992 a | 0.913 a |
| | 7 | 0.945 a | 1.852 b | 1.876 b | 2.003 b | 1.510 a | 1.331 a | 1.620 a | 1.665 a | 1.728 a | 1.687 a | 1.254 a | 1.093 a | 1.880 b | 1.577 a | 1.325 a |
| | 30 | 1.124 a | 1.844 b | 1.344 a | 1.904 b | 1.377 a | 1.138 a | 0.702 a | 1.276 a | 1.275 a | 1.557 b | 1.337 a | 1.268 a | 1.335 a | 1.588 a | 1.585 a |
| Phenolic compounds | 1 | 0.896 a | 0.006 b | 0.847 a | 0.901 a | 0.678 a | 0.957 a | 0.000 b | 0.637 a | 0.797 a | 0.757 a | 0.820 a | 0.063 b | 1.173 a | 0.510 a | 0.888 a |
| | 7 | 0.313 a | 1.142 b | 0.995 b | 1.077 b | 1.864 c | 0.658 a | 1.676 b | 1.421 b | 1.576 b | 1.917 b | 0.952 a | 0.593 a | 0.881 a | 1.470 a | 1.451 a |
| | 30 | 0.886 a | 1.605 b | 1.057 a | 2.184 c | 1.836 b | 0.922 a | 0.761 a | 0.966 a | 0.881 a | 1.683 b | 1.432 a | 0.799 b | 1.408 a | 1.171 a | 0.985 a |
| Polymers | 1 | 2.182 a | 0.015 b | 1.929 a | 1.878 a | 1.621 c | 1.653 a | 0.016 b | 1.632 a | 1.643 a | 1.845 a | 1.385 a | 0.040 b | 1.378 a | 1.516 a | 1.394 a |
| | 7 | 1.522 a | 2.267 b | 2.206 b | 2.240 b | 1.848 ab | 1.295 a | 1.848 ab | 1.868 ab | 1.451 a | 2.002 b | 1.806 a | 1.595 a | 2.243 b | 1.985 a | 1.818 a |
| | 30 | 1.550 a | 2.005 b | 2.042 b | 2.133 b | 2.227 b | 1.310 a | 1.107 a | 1.348 a | 1.322 a | 1.933 b | 2.033 a | 1.880 a | 2.086 a | 1.908 a | 2.069 a |

**Table 4.** Fungal catabolic activity of self- or cross-inoculated soils during a 30-d incubation experiment. Heat-sterilized and non-inoculated soils (Val\*, Rom\*, Vic\*) together with non-sterilized and non-reinoculated soils (Val, Rom, Vic) were taken as control treatments. At each sampling time, different letters indicate a significant ($p < 0.05$) difference among means (n = 3).

| Substrate Category | Time (days) | Soil Treatments | | | | | | | | | | | | | | |
| --- | --- | --- | --- | --- | --- | --- | --- | --- | --- | --- | --- | --- | --- | --- | --- | --- |
| | | Vallombrosa | | | | | Romola | | | | | Vicarello | | | | |
| | | Val | Val* | Val*+Val | Val*+Rom | Val*+Vic | Rom | Rom* | Rom*+Rom | Rom*+Val | Rom*+Vic | Vic | Vic* | Vic*+Vic | Vic*+Val | Vic*+Rom |
| Amines | 1 | 1.104 a | 0.025 b | 1.493 a | 1.806 c | 1.064 a | 1.116 a | 0.340 b | 1.388 a | 0.563 ab | 1.614 c | 1.314 a | 0.001 b | 1.074 a | 1.410 a | 1.132 a |
| | 7 | 1.841 a | 1.301 a | 1.409 a | 1.347 a | 1.352 a | 1.111 a | 2.013 b | 1.591 a | 1.323 a | 1.197 a | 1.632 a | 1.396 a | 1.286 a | 1.381 a | 1.323 a |
| | 30 | 0.486 a | 0.869 a | 0.593 a | 1.201 b | 1.592 c | 1.466 a | 0.177 b | 1.243 a | 0.769 a | 0.670 a | 1.366 a | 1.683 a | 1.374 a | 1.677 a | 1.762 a |
| Amino acids | 1 | 0.894 a | 0.006 b | 1.018 a | 1.329 c | 1.154 a | 1.081 a | 0.322 b | 1.001 a | 1.064 a | 1.200 a | 1.233 a | 0.000 b | 0.995 a | 0.885 a | 1.025 a |
| | 7 | 0.987 a | 0.994 a | 1.146 a | 1.332 a | 1.187 a | 1.230 a | 1.293 a | 1.361 a | 1.135 a | 1.393 a | 1.198 a | 1.138 a | 0.974 a | 1.214 a | 1.181 a |
| | 30 | 0.657 a | 1.058 a | 0.996 a | 1.164 a | 1.257 b | 1.035 a | 0.294 b | 1.157 a | 0.729 a | 1.236 a | 0.911 a | 1.364 a | 1.263 a | 1.306 a | 1.307 a |
| Carbohydrates | 1 | 0.900 a | 0.032 a | 0.902 a | 1.181 a | 0.971 a | 0.819 a | 0.235 a | 0.931 a | 0.869 a | 1.172 a | 1.105 a | 0.002 b | 0.927 a | 0.797 a | 0.892 a |
| | 7 | 0.853 a | 0.974 a | 0.986 a | 1.327 b | 1.167 a | 1.243 a | 1.417 a | 1.317 a | 1.222 a | 1.371 a | 1.158 a | 1.175 a | 1.155 a | 1.177 a | 1.180 a |
| | 30 | 0.515 a | 0.993 ab | 0.796 a | 1.075 ab | 1.174 b | 0.930 a | 0.165 b | 1.204 a | 1.069 a | 1.273 a | 0.750 a | 1.249 a | 1.168 a | 1.061 a | 1.077 a |
| Carboxylic acids | 1 | 1.220 a | 0.027 b | 1.200 a | 1.357 a | 1.061 a | 0.851 a | 0.353 b | 1.190 a | 1.083 a | 1.362 b | 1.183 a | 0.000 b | 1.100 a | 1.000 a | 0.996 a |
| | 7 | 1.011 a | 1.051 a | 1.132 a | 1.390 a | 1.283 a | 1.284 a | 1.503 a | 1.378 a | 1.336 a | 1.376 a | 1.255 a | 1.131 a | 1.138 a | 1.180 a | 1.214 a |
| | 30 | 0.797 a | 1.001 a | 1.049 a | 1.136 ab | 1.232 b | 1.086 a | 0.251 b | 1.327 a | 1.258 a | 1.532 b | 1.092 a | 1.364 a | 1.260 a | 1.382 a | 1.366 a |
| Nucleotides | 1 | 1.246 a | 0.024 b | 1.231 a | 1.580 c | 1.080 a | 1.242 a | 0.425 b | 1.045 a | 1.165 a | 1.178 a | 0.824 a | 0.000 b | 1.129 a | 0.992 a | 1.178 a |
| | 7 | 0.761 a | 0.751 a | 0.851 a | 0.995 a | 1.041 a | 1.468 a | 1.577 a | 1.448 a | 1.408 a | 1.446 a | 1.428 a | 0.977 a | 1.215 a | 1.182 a | 1.160 a |
| | 30 | 0.802 a | 0.927 a | 1.000 a | 0.933 a | 0.949 a | 1.081 a | 0.586 a | 1.468 a | 0.596 a | 0.812 a | 0.900 a | 1.389 a | 1.308 a | 1.542 a | 1.561 a |
| Polymers | 1 | 0.900 a | 0.007 b | 0.765 a | 1.233 c | 0.681 a | 0.558 a | 0.119 a | 0.734 a | 0.452 a | 1.076 a | 0.778 a | 0.000 b | 0.563 a | 0.599 a | 0.724 a |
| | 7 | 0.652 a | 0.708 a | 0.740 a | 1.357 b | 1.061 a | 1.005 a | 1.313 a | 1.208 a | 1.042 a | 1.424 a | 0.920 a | 1.065 a | 1.131 a | 1.105 a | 1.166 a |
| | 30 | 0.178 a | 0.678 a | 0.459 a | 0.992 a | 1.133 b | 0.714 a | 0.137 b | 1.050 a | 0.835 a | 1.271 a | 0.488 a | 0.831 a | 0.863 a | 0.917 a | 0.923 a |

Remarkably, the sterilized Val* soils displayed similar values. On the other hand, fungal catabolic activity was less affected by treatments than bacteria and showed more constant values over time across the different soils. Interestingly, at the end of the incubation time, the fungal AWCD values were higher in sterilized soils than in pristine untreated soil. In contrast, bacterial AWCD values of sterilized soils were lower than the untreated control, except for Val*, which exhibited higher values.

### 3.3.1. Community-Level Physiological Profile of the Vallombrosa Soil

Soon after the heat sterilization (day 1), both bacterial and fungal AWCD values were almost zero in Val* soils. However, with elapsing time they returned to levels comparable to or higher than those of the non-autoclaved pristine soil (Figure 2). In Val*+Val and Val*+Rom soils, the AWCD values of both bacterial and fungal communities followed similar trends and were not significantly different from those of the Val soil ($p < 0.05$, see Supplementary Materials). In contrast, for the Val*+Vic soil the bacterial AWCD was significantly lower and the fungal AWCD significantly higher as compared to the control soil. Notably, after 7 days of incubation, the AWCD values for bacterial communities were higher in the sterilized and all reinoculated soils than in the Val soil, whereas the AWCD values of the fungal communities were significantly higher than the control soil only in the Val*+Vic treatment. After 30 days of incubation, the AWCD values were significantly higher in the sterilized and all reinoculated soils than the control soil for both the bacterial and fungal communities.

Grouping the results as substrate utilization highlighted the increase of the bacterial catabolic activity of Val* soil after 7 and 30 days compared to its initial value. Interestingly, at the end of the incubation period, Val* showed the highest catabolic activity with amines and carbohydrate substrates (Table 3). On the other hand, Val*+Rom showed the highest values with amino acids, carboxylic acids and phenolic compounds, whereas Val*+Vic showed the highest catabolic activity on polymer substrates.

Regarding the fungal substrate utilization (Table 4), results showed relevant catabolic activity in Val* since the beginning of the experiment, with values similar to the other samples. At day 1, Val*+Rom samples displayed the highest values of catabolic potential on all the substrates compared to the other samples. On the other hand, at day 30 Val*+Vic showed the highest values on all the substrates except for nucleotides, which displayed the highest value in Val*+Val soils.

### 3.3.2. Community-Level Physiological Profile of the Romola Soil

As already observed in Val soils, autoclaving soils rapidly annihilated microbial physiological activity, especially in bacterial community (Figure 2). However, after 7 and 30 days of incubation the sterilized soils returned to values comparable to the non-sterile soils, with the AWCD values of Rom* being significantly higher than those of the Rom soil after 7 days and lower after 30 days both for bacteria and fungi. After 7 days of incubation, the AWCD values for bacterial communities were significantly higher in all sterilized and reinoculated soils compared to the pristine control soil (Rom), whereas the AWCD values of the fungal communities were significantly lower than the control soil only in Rom*+Val soil. After 30 days of incubation, the bacterial AWCD values of Rom*+Rom and Rom*+Val were similar to those of Rom soil, whereas Rom*+Vic showed significantly higher readings. The AWCD values of fungal communities after 30 days of incubation were significantly higher for the Rom*+Rom and Rom*+Vic soils as compared to that of the Rom soil.

The substrate utilization of Romola soils highlighted the higher values of the bacterial catabolic activity of the Rom*+Vic soil after 30 days compared to the other soils. Interestingly, at the beginning of the incubation period, Rom*+Vic showed the highest catabolic activity with all the substrates, except for carboxylic acids and phenolic compounds, which were higher in Rom soil (Table 3). As observed with Val soils, after 7 and 30 days, the sterilized Rom* samples showed values comparable with the untreated and reinoculated soils. Regarding the fungal substrate utilization (Table 4), results confirmed Rom*+Vic as

the soils with the highest catabolic values at the beginning of the experiment (day1) with all the substrates except for nucleotides. Moreover, after 30 days, they also displayed the highest values in all the substrate categories except for amines and nucleotides.

### 3.3.3. Community-Level Physiological Profile of the Vicarello Soil

Once again, significant differences were found in the AWCD values of bacterial communities between Vic and Vic* soils, whereas those of the fungal communities of sterilized and inoculated soils showed significantly lower AWCD values than Vic soil (Figure 2). After 7 days of incubation, the AWCD values for bacterial communities were significantly lower in Vic* and significantly higher in the Vic*+Val and Vic*+Vic than in Vic treatments, whereas no significant differences were found in the AWCD values of the soil fungal communities. After 30 days of incubation, the AWCD values of the bacterial communities were significantly lower in the Vic* soil as compared to the other treatments, whereas the AWCD of fungal communities reached values significantly higher in the sterilized and all reinoculated soils as compared to Vic.

The bacterial substrate utilization of Vic soils highlighted the increase of the bacterial catabolic activity of phenolic compounds and polymers in Vic*+Vic. Sterilized Vic* after 7 and 30 days showed values similar to the untreated control and the inoculated soils. Overall, there were no significant differences among all the treatments with Vic soils. On the other hand, at day 1 the fungal catabolic activity showed the highest values in the untreated control Vic with all the substrates except for amines and nucleotides. The sterilized Vic* soil showed the highest values with amines, amino acid and carbohydrates substrates whereas Vic*+Rom provided the highest catabolic values after 30 days.

### 3.3.4. Multivariate Analysis of CLPP Data

The multivariate analysis (NMDS) performed by means of the Manhattan distance of different substrate pattern consumption is reported in Figure 3.

Looking at the bacterial communities in Romola, as expected at the beginning, all the sterilized soils were grouped separately from the non-sterile soils. After 7 days, the different soil groups clustered separately, with Rom*+Val very close to Rom* soils. After 30 days, all the soils grouped together again, regardless of the different inocula, except for Rom*+Vic, which clustered separately. Rom*+Val was the most similar to the control Rom treatment. On the other hand, the AWCD values of the fungal community in sterilized Romola soils were similar at the beginning of the experiment and after 30 days but not after 7 days, when they changed completely. Overall, after 7 days most of the inoculated soils were more similar to autoclaved samples than to the Rom-inoculated soils, especially Rom*+Rom. Interestingly, autoclaved soils clustered very close to Rom*+Rom. At the end of the incubation, such differences were confirmed for all the samples except for sterile control soils and with Rom*+Val as the closest group to Rom control soils.

Overall, bacterial CLPP data from Vallombrosa soils were much more similar to each other than in Romola. After 1 day of incubation, both the sterile soils and all the reinoculated soils except for Val*+Vic clustered together, whereas the untreated and sterile controls clustered separately. After 7 days, all the inoculated soils clustered together, except for Val*+Val, whereas the untreated Val soils clustered separately again. Val* displayed similar results to Val*+Rom and Val*+Vic. After 30 days of incubation, Val* soils and all the inoculated soils clustered together, whereas the Val control clustered separately. The fungal CLPP results showed that after 1 day of incubation the sterilized soils clustered separately from all the other samples. More specifically, the Val* soils clustered together with Val*+Val samples and close to Val*+Rom, whereas Val*+Vic and the untreated Val soils clustered separately. Conversely, after 30 days, all the samples were grouped in a unique wide cluster, showing a high variability also among replicates.

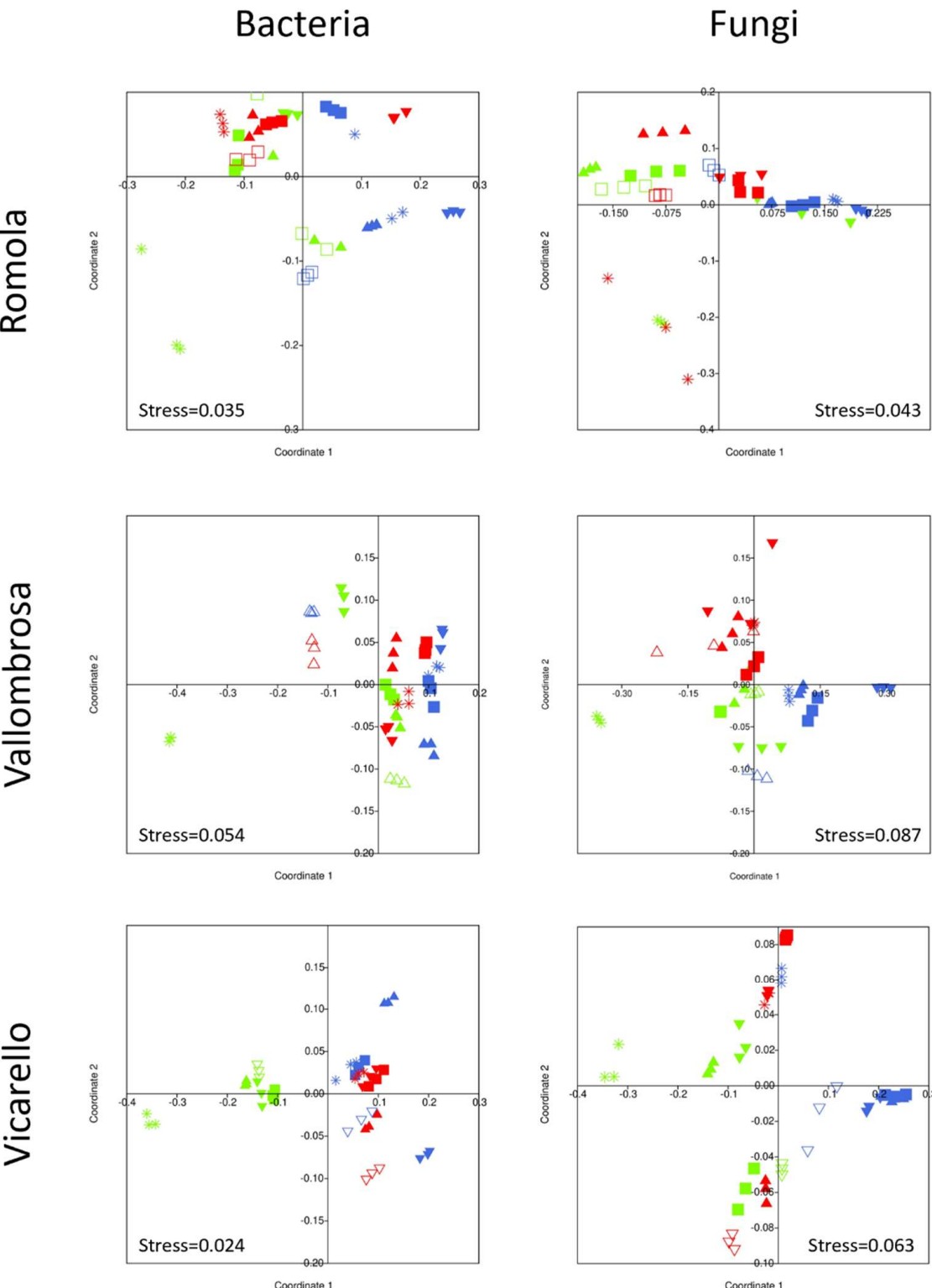

**Figure 3.** NMDS of the CLPP data obtained from the Biolog Ecoplates (bacteria) and FF plates (fungi) from Romola (square), Vallombrosa (triangle) and Vicarello (triangle) soils and sterilized soils (star) as well as self- or cross-reinoculated with Romola (filled square), Vallombrosa (filled triangle) and Vicarello (filled inverted triangle) soils after 1 day (green), 7 days (blue) and 30 days of incubation (red).

After 1 day of incubation, the bacterial CLPP data showed that Vicarello samples were highly distinct after 7 and 30 days (Figure 3), and sterile soils clustered separately. After 7 days, Vic*+Val and Vic*+Vic exhibited the most relevant differences compared to the other samples, which clustered together, including the controls. However, after 30 days all the samples seemed to be displayed into the same cluster, except for the Vic soils, which grouped separately.

Fungal CLPP values, in contrast, were much more variable: at the beginning of the experiment all the sterilized soils formed a separate cluster on the left, whereas the untreated Vic soils clustered with Vic*+Rom. After 7 days, all the inoculated soils clustered together, whereas both sterile and non-sterile controls clustered separately. Then, after 30 days all the soil groups seemed to cluster separately, except for the sterile soil which was very similar to Vic*+Vic soils.

### 3.4. Bacterial and Fungal Community Structure

The PCR amplifications of bacterial 16S and fungal 18S rRNA genes showed a good reproducibility for all soils. Autoclaving completely degraded nucleic acids, as after one day of incubation only one Rom* sterilized non-inoculated soil replicate showed a faint 16S rDNA amplification signal, but no amplification signal was obtained for 18S rDNA. For sterilized non-inoculated soils incubated for 7 and 30 days, PCR amplification of bacterial 16S rDNA was observed in one replicate of the Rom* soil and in all three replicates of Vic* (after 30 days), whereas PCR amplification of fungal 18S rDNA was observed only for one replicate of Vic* autoclaved soil (after 30 days) (Figure S2).

The bacterial DGGE profiles of the control soils showed a greater complexity than those of the fungal ones, and both bacterial and fungal communities of all control soils showed a high stability during the 30-day incubation period. The sterilized inoculated soils showed all distinct DGGE banding patterns with shifts in the number, distribution and intensity of the banding profiles, indicating the establishment of different microbial communities during the incubation period (Figure 4), and in sterilized inoculated soils, changes in the bacterial DGGE profiles were greater than those of fungal communities (Figure 4).

The NMDS analysis conducted on DGGE profiles of non-sterilized control soils showed that the composition of both bacterial and fungal communities from Val, Rom and Vic pristine soils displayed high similarity for the entire incubation period (Figure 4). Variation in bacterial and fungal community composition of control soils were significant, as revealed by both ANOSIM (R = 1, $p < 0.01$ for 16S-DGGE; R = 1, $p < 0.01$ for 18S-DGGE) and PERMANOVA (F = 26.27, $p < 0.01$ for 16S-DGGE; F = 10.86, $p < 0.01$ for 18S-DGGE) tests.

NMDS analysis conducted on sterilized self- and cross-reinoculated soils showed that at the beginning of the incubation period the composition of microbial communities was, in general, more similar to that of the pristine soil used for reinoculation than to the other soils. Successively, the composition of both bacterial and fungal communities of reinoculated soils evolved with time, although a broad overlapping grouping could be highlighted depending on the soil used for reinoculation (Figure 4). In all cases, ANOSIM and PERMANOVA analysis confirmed statistically significant evidence in NMDS cluster patterns with R values ranging between 0.433 and 0.955 ($p < 0.001$) and F values ranging between 3.261 and 9.846 ($p < 0.001$), respectively.

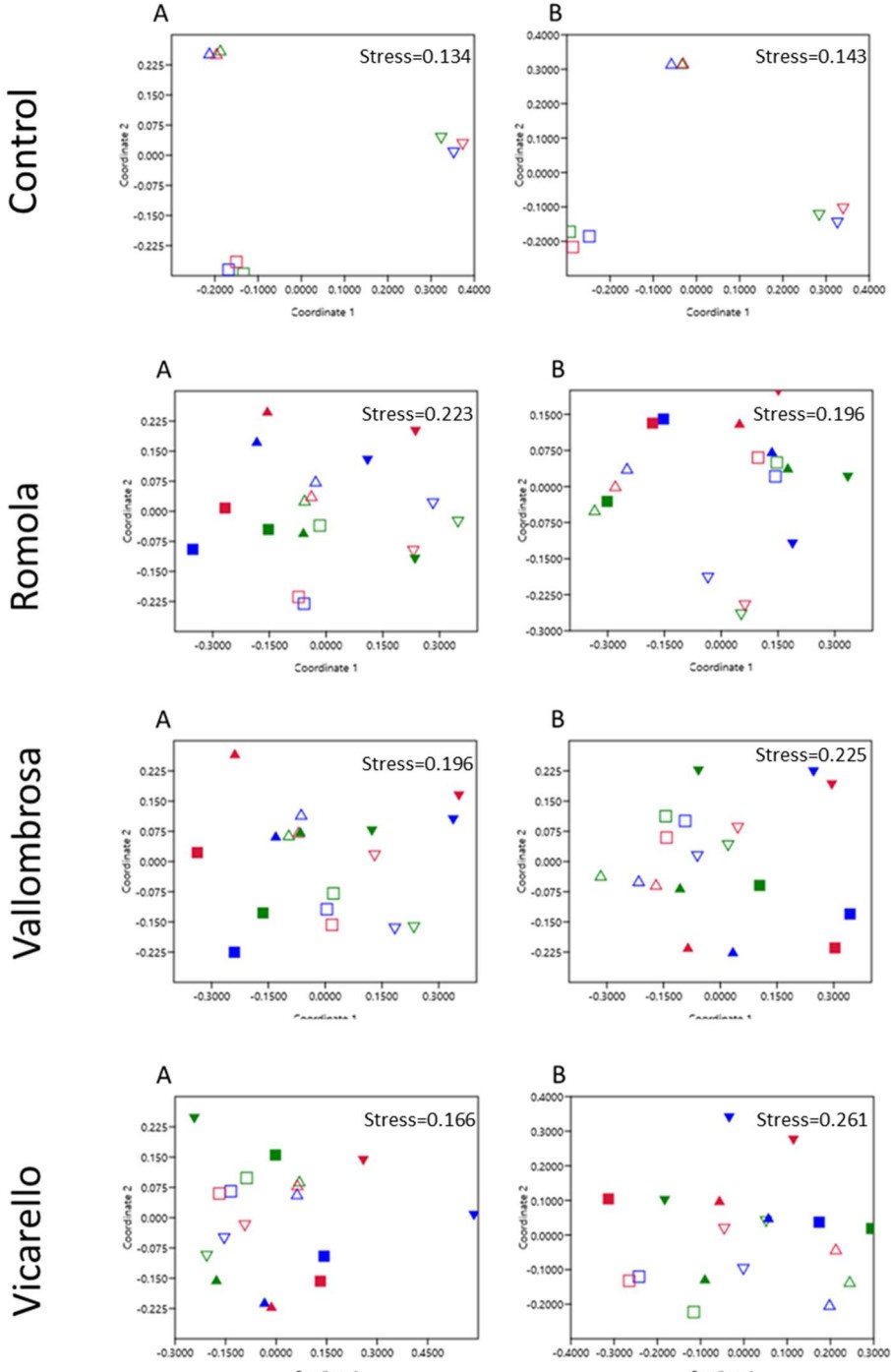

**Figure 4.** NMDS ordination plots of 16S (**A**) and 18S (**B**) rRNA gene DGGE profiles obtained from Vallombrosa (empty triangle), Vicarello (empty inverted triangle) and Romola (empty square) control soils and from sterilized soils reinoculated with Vallombrosa (filled triangle), Vicarello (filled inverted triangle) and Romola (filled square) soils, incubated for 1 (green), 7 (blue) and 30 days (red).

### 3.5. Linking Enzyme, Biochemical and Metabolic Data

In order to better relate the enzymatic, biochemical, catabolic and molecular results, we performed a pairwise Pearson's correlation analysis to measure the statistical relationship, or association, between the different functional variables. In Figure 5, the significant positive (blue) and negative (red) correlations were displayed (all the correlation and *p*-values are reported in Table S1). Most of the correlations were positive, except for fungal

's' value (s_FF), indicating the time required to reach the asymptotic increase of the AWCD curve. It was negatively correlated to alkaline phosphatase, dsDNA, ATP and cumulative respiration (Ccum) and with the kinetic parameters of bacterial catabolic activity 's_B' and CV_B. Alkaline phosphatase is positively correlated to most of the catabolic activity provided by both bacteria and fungi. However, acid phosphatase and β-glucosidase were positively related only to bacterial catabolism and ammonification. Protease and urease did not show any significant correlation with microbial catabolic activity but with the microbial diversity indices (H' and richness). As expected, most of the microbial catabolic variables were positively correlated.

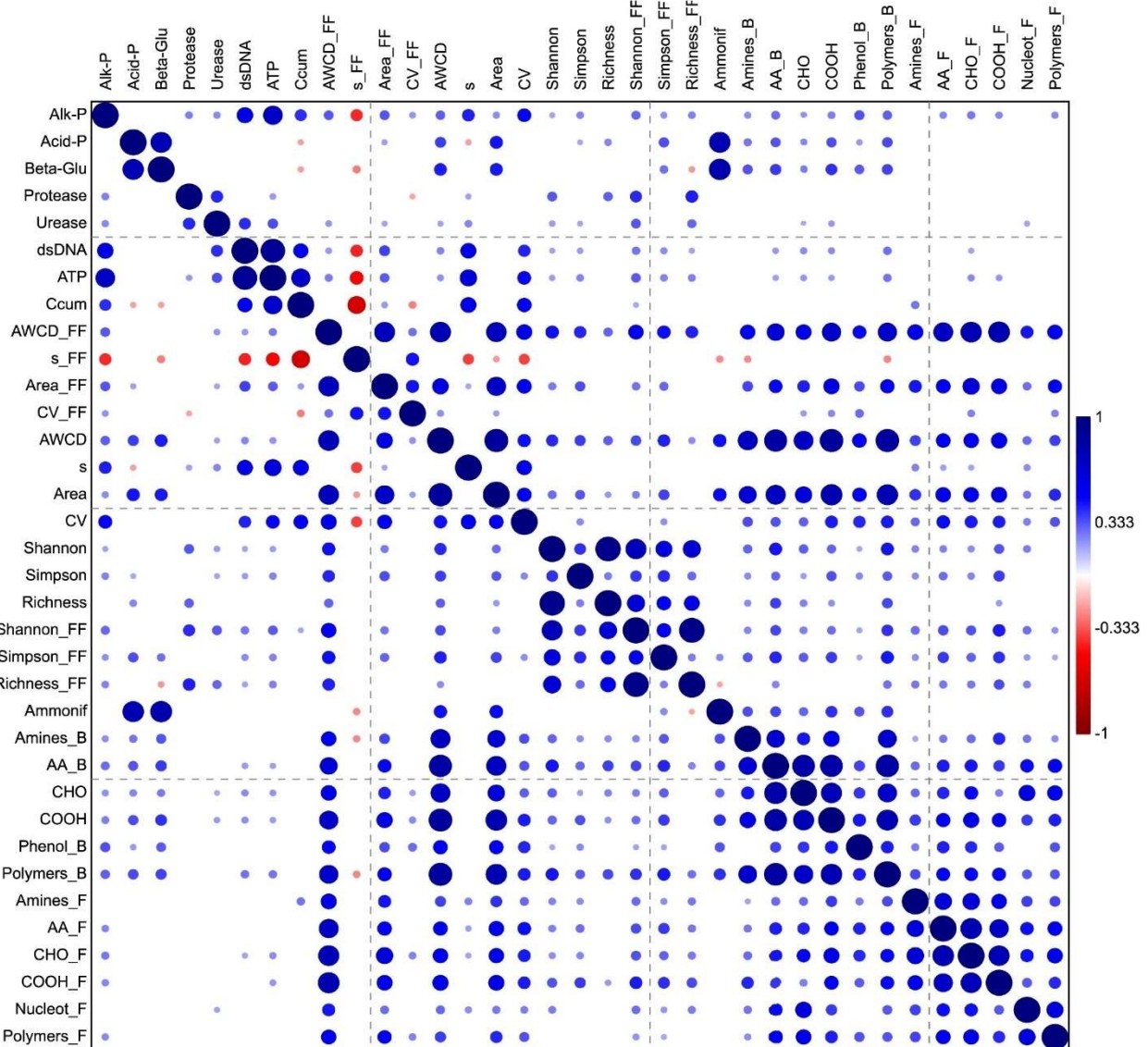

**Figure 5.** Pairwise Pearson's correlation analysis. The two specular triangles display the Pearson's correlation coefficient (r) between each of the two soil characteristics. Blue and red colors indicate positive and negative correlations, respectively. The color density and the square size reflect the scale of the correlation. Color density and circle size demonstrate the significance level, and *p*-values above 0.05 were regarded as insignificant and labeled in white color. Variables indicated with B or FF are referred to bacteria or fungi, respectively. Abbreviations: Alk-P (alkaline phosphatase), Acid-P (acid phosphatase),



Beta-Glu (beta-glucosidase), dsDNA (double-strand DNA), Ccum (cumulative respiration), s_(catabolic kinetics), Area (area of the CLPP curve), CV (catabolic versatility), Ammonif (ammonification), AA (amino acids), CHO (carbohydrates), COOH (carboxylic acids), Phenol (phenolic compounds), Nucleot (nucleotides).

The principal component analysis (PCA) was carried out on the same variables, including enzyme, biochemical, metabolic and molecular data collected from Val, Rom and Vic soil with all the subsequent sterilized and re- or cross-inoculated soils over time (1, 7 and 30 days) to reveal their effects on the relative distribution of the soils (Figure 6).

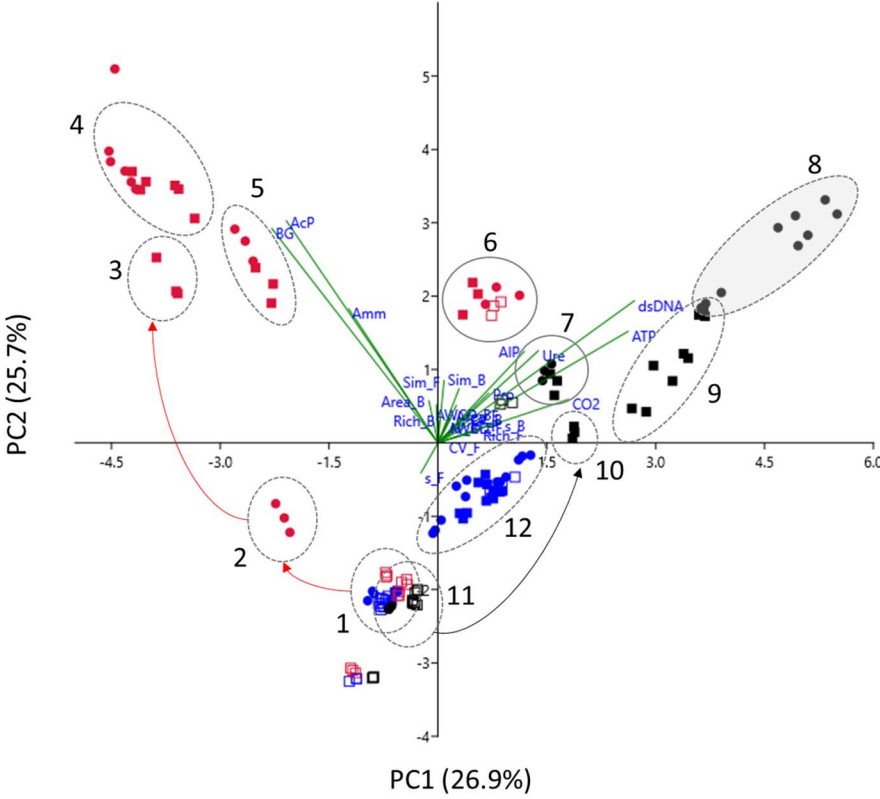

**Figure 6.** Principal component analysis (PCA) score plots based on the relative distribution of the original soils Val (red), Rom (blue) and Vic (black) together with sterilized and non-inoculated soils, Val*, Rom* and Vic*, as well as the sterilized and self- or cross-reinoculated soils after 1 day (empty squares), 7 days (circles) and 30 days (full square) of incubation. Different clusters have been numbered as follows: 1 (Val*, day1), 2 (Val*, day 7), 3 (Val*, day30), 4 (Val*+Val, Val*+Rom), 5 (Val*+Vic), 6 (Val), 7 (Vic), 8 (Vic*+Val, Vic*+Vic, Vic*+Rom, day 7), 9 (Vic*+Val, Vic*+Vic, Vic*+Rom, day 30), 10 (Vic*, day30), 11 (Vic*, day 1 and 7), 12 (Rom, Rom*, Rom*+Rom, Rom*+Val, Rom*+Vic). Percentages correspond to the variance explained in each axis (PC1 = 26.9%, PC2 = 25.7%).

The outcomes of the PCA showed a clear separation among Val (red), Rom (blue) and Vic (black) soils along PC1 and PC2, which explained, respectively, 26.9% and 25.7% of variance (Figure 6). The results also indicated contrasting contributions to PC1 by β-glucosidase, acid phosphatase and ammonification (negative loadings) and alkaline phosphatase, urease and most of the biochemical properties (positive loadings). The contribution of microbial catabolic activity and diversity appeared to be less relevant. Interestingly, bacterial catabolic efficiency (Area_B) and taxonomical richness (Rich_B) were closely associated with β-glucosidase, acid phosphatase and ammonification and positively related to Val soils, whereas fungal catabolic potential (AWCD_FF) and taxonomic richness (Rich_F) were associated with urease, alkaline phosphatase (AlP) and soil respiration (Ccum), as well as dsDNA and ATP, and were strongly related to Vic soils.

All the samples at day 1 clustered together below the PC1 axis (empty symbols). It is worth noting also that Rom soils clustered along the negative values of the PC2 axis, regardless of the sampling time or the reinoculated soil (cluster 12). In contrast, Vic and Val soils were positively related to PC2 but contrastingly related to PC1. In fact, Val soils are mainly distributed along the negative PC1 values, whereas Vic soils are distributed along the positive PC1 values. Val* soils showed a relevant resilience over time, clustering progressively closer to inoculated Val soils 7 days (clusters 2) and 30 days (cluster 3) after sterilization. Val*+Val and Val*+Rom soils clustered together after 7 and 30 days, whereas Val*+Vic clustered separately (cluster 5). Both original Val and Vic soils grouped separately (clusters 6 and 7, respectively). Interestingly, inoculated Vic soils grouped according to their temporal changes rather than to the treatment. In fact, Vic*+Vic, Vic*+Rom and Vic*+Val clustered together after 7 days (cluster 8) and 30 days (cluster 9). Finally, sterilized Vic* soils after day 1 and day 7 grouped together (cluster 11) close to cluster 1. However, after 30 days, Vic* clustered close to the untreated Vic samples (cluster 10). In contrast, all the Rom soils were poorly affected by time and treatments, clustering together under the PC1 axis (cluster 12), showing the lowest biological and biochemical activity.

## 4. Discussion

### 4.1. Soil Respiration, N Ammonification, ATP and Microbial Biomass

In general, cumulative respiration was significantly higher in all the sterilized and inoculated soils than in non-inoculated soils (Figure 1). These results paralleled those reported by Powlson and Jenkinson [15] after soil biocidal treatments, a phenomenon generally ascribed to the increase of easily mineralizable organic substrates made available to microorganisms after soil autoclaving, including the microbial biomass C. As expected, Rom soils showed the lowest values, compared to Val and Vic. However, it cannot be excluded that the large $CO_2$ flush after soil autoclaving could also result from a priming effect [44].

The sterilized soils, inoculated or not, showed higher ammonification rates than untreated soils (Figure 2). Though ammonification in soil generally releases only a small fraction of total soil N (in the order of 1%–1‰) and does not significantly alter the composition of total organic N, the released $NH_4$+-N can be directly assimilated by the actively growing microorganisms or undergo nitrification and ammonia volatilization independently of the soil pH value [45,46]. Remarkably, the result showed that the $CO_2$-C and $NH_4^+$-N flushes after autoclaving were relatively independent of the source of the inoculum and different in magnitude not in trends. We explain this finding assuming that the SOM mineralization during the early stages of soil recolonization was more likely influenced by the organic C quality and availability than by the composition and biomass of the microbial communities. These results support the 'abiotic gate' hypothesis of the predominance of abiotic over biotic factors in the control of SOM mineralization [47]. Interestingly, while the ammonification rates of the autoclaved non-reinoculated Rom* and Vic* soils remained similar to those of the control soils (Rom, Vic) throughout the incubation period, in the Val* soil it was significantly higher than in the untreated Val soil. This result can be related to the high urease activity already found after 7 days of incubation in the Val* soil, different from the Rom* and Vic* soils (Table 2). Urease activity is a rate-limiting step in N ammonification, and the studied Val soil displayed high urease potential activity, also, when inoculated in the neutral and alkaline soils (Table 2).

Results of soil ATP content showed that the Rom* and Vic* soils were recolonized by indigenous or exogenous microbial communities after 30 days of incubation, whereas in Val* soil the ATP and dsDNA content levelled off to ca. 30% of the initial value, regardless of the inoculation source (Figures 2c and S1). The increasing amount of microbial biomass from Val to Rom and Vic sterilized and reinoculated soils, is likely due more to their pH values than organic C content. Accordingly, lower microbial proliferation in acidic as compared to neutral and alkaline soils has been previously reported [48], also after soil biocidal treatments [49].

*4.2. Soil Enzymatic Activities*

Overall, soil autoclaving reduced hydrolase activity to undetectable levels in sterilized non-inoculated soils, whereas the low enzyme activity levels detected in the sterilized inoculated soils at early stages reflected the enzyme activity of the inocula (Table 2). The increase of the soil enzymatic activities in all the sterilized (whether reinoculated or not) soils after 7 or 30 days of incubation showed, to varying extents, the recovering capacity of native or exogenous communities to synthesize new enzymes. Moreover, by comparing the hydrolase values, it was possible to discriminate the contribution of the recovering native microbial communities of each inoculated soil by the hydrolase activity of the sterilized soils, which was generally greater after 7 than 30 days of incubation for most of the soils. Studying the production and persistence of the acid and alkaline phosphomonoesterase activities in the same soils, Renella et al. [50] reported that under steady conditions acid phosphomonoesterase activity predominated in the Vallombosa acidic soil, whereas alkaline phosphomonoesterase activity predominated in the Romola neutral and Vicarello alkaline soils, but during the microbial growth induced by the incorporation of plant litter alkaline phosphomonoesterase was produced more in the Vallombrosa soil, whereas acid phosphomonoesterase was produced more in the Vicarello alkaline soil.

The peak of β-glucosidase activity observed during the most active microbial growth phase (day 7) could be due to the large N and P availability. A positive correlation between β-glucosidase and active soil microorganisms was reported by Knight and Dick [51], and stimulation of β-glucosidase activity by the availability of N and P has been previously reported [52,53]. Overall, the adopted experimental approach showed that microbial communities originating from soils with different physicochemical properties have the potential to produce hydrolitic activities at far higher levels than in the native soils under steady conditions. Remarkably, β-glucosidase activity dramatically increased in Val sterilized and inoculated soils compared to the untreated control. In contrast, in Rom sterilized and inoculated soils, β-glucosidase displayed similar values to the control, whereas it decreased in Vic sterilized and inoculated soils. This result contrasts with a study by Stark et al. [54], which highlighted that in nutrient-poor and acidic tundra, increased nutrient availability and pH reduced β-glucosidase activity, while it had no effects on the total bacterial or fungal biomass. The reasons for this are still uncertain but might reflect the altered stoichiometry of microbial nutrient demands and accessibility to soil C substrates after sterilization [55].

The results of microbial biomass (dsDNA) (Figure S1) and CLPP (Figure 2, Tables 3 and 4) of the inoculated soils suggested that microorganisms originating from different soils can degrade the native organic matter during the recolonization process of soils having physico-chemical properties highly different from the original soils, thus supporting the hypothesis that soil colonization can be due to members of native microbial communities surviving soil perturbation, but could also be due to allochthonous microorganisms entering the soil during the perturbation event [56]. In support of this, Pettersson and Bååth [57] went on to show that bacterial communities exert a better colonizing ability when dominant microbial members are removed by chloroform fumigation. The availability of low-molecular weight organic C along with the reduced microbial competition in the habitat are possibly factors that prevent the organic C limitation typical of soils under steady conditions, including C-rich forest soils [58]. Values of ATP content also showed that significantly higher biomass could be hosted by Romola sandy neutral and Vicarello clay alkaline soils inoculated with exogenous microbial communities as compared to the untreated soils, showing that the C-limiting conditions as well as the microbial interactions kept the soils at biomass levels below their maximum carrying capacity (Figure 3). This was not true for the Vallombrosa soil, in which the acidic pH value could likely represent a limiting factor on microbial proliferation. In fact, microbial communities adapt to soil pH through selection and synthesis of many osmotically active metabolites, membrane proteins and ionic transporters [59]. Moreover, bacterial communities are more sensitive to acidification than fungal communities as compared to neutral sub-alkaline pH values [60,61]. This has been confirmed, also,

by our results obtained with the Biolog microplate approach. The lack of recovery of urease and protease activity to the original levels could be likely due to either high availability of N-rich metabolites in the sterilized soils or to the low synthetic capacity of the recolonizing microbial populations.

Overall, our results confirmed that enzymatic activities are strongly expressed during the community colonization of sterilized soils and they are likely key factors for the rapid adaptation shown by the microbial communities in soils, in agreement with Hoshino and Matsumoto [62], who reported that microbial activity is a more key feature than microbial diversity during the early stages of soil recolonization.

### 4.3. Soil Community-Level Physiological Profile

Despite the well-known limitations of the Biolog microplate approach [63], the CLPP assessment has been largely used for assessing microbial functional diversity and the combined use of ECO and FF microplates has been recently proposed as a functional biodiversity indicator of the soil microbial communities [64,65]. The microbial catabolic activity described in this paper is based on all carbon sources and on grouped sources defined as amines/amides, amino acids, carbohydrates, carboxylic acids, polymers and nucleotides or other substrates, as previously proposed by other authors [66,67], determining to make bacterial and fungal results comparable.

The highest values of bacterial catabolic potential were observed in Val soils, whereas sterilized and reinoculated soils displayed higher AWCD values than the untreated control (Figure 2). Our result indicates that the catabolic activity of the microbial communities increased under conditions of high nutrient availability, like those created by sterilization, in accordance with previous studies [23,68]. This result also highlights the importance of soil physicochemical attributes and local abiotic conditions on the shaping of functional traits of microbial communities. In fact, the overall catabolic activity of fungal communities of Val soils inoculated with more alkaline soils, such as Rom and Vic, showed higher values than the untreated control, suggesting an efficient response of the native fungal community to the exogenous input. In contrast, the bacterial catabolic activity detected on most of the substrate categories just after the inoculation (day 1) was higher in the untreated Val soils than in the others but, after 7 and 30 days, the catabolic activity of the inoculated soils was similar or even higher than Val. This result indicates the longer time required by bacteria to adapt their metabolism to the new environmental conditions in an acidic soil compared to fungi, as mentioned in the previous section. This indication seems to be also confirmed by the sterilized soils, where bacteria showed a low or no catabolic activity at day 1, which increased after 7 and 30 days, whereas fungal communities were constantly active since day 1, providing similar values up to day 30. Moreover, depending on the substrate category, different catabolic rates have been observed between Val*+Rom and Val*+Vic samples, especially for fungi (Tables 3 and 4). For example, amines are mostly used by fungi at day 1 in Val*+Rom soils and their use decreases after 7 and 30 days. In contrast, Val*+Vic displayed increasing values from day 1 up to day 30 when the highest catabolic activity was observed. Similar catabolic activity was observed for amino acids, carbohydrates, carboxylic acids and polymers. These observations are apparently not in accordance with the results reported by other authors for a similar soil. For instance, a study carried out on a sandy soil with pH 5.2 reported that the soil fungal community mostly used carboxylic acids and amino acids for the first 90 days and carbohydrates as well as polymers after 270 days [69]. However, soil functional diversity is strongly connected with the need to adapt to the environment and several soil features other than pH might shape the use of carbon sources located in the FF microplates. For example, the catabolic activity of both bacterial and fungal communities is generally higher in neutral Rom soils inoculated with the alkaline Vic soil (Rom*+Vic) than in those inoculated with the acidic Val soil (Rom*+Val), regardless of its higher content of organic carbon, suggesting that pH is not the only driver shaping microbial functional diversity, confirming the findings of previous works [70,71].

Remarkably, the kinetics of the fungal catabolic activity highlighted by the s value are negatively correlated to the bacterial metabolic rate and catabolic versatility (Figure 5), thus confirming that, despite the clear importance of edaphic factors in controlling microbial communities, fungal/bacterial interactions play a major, although causally unclear, role in shaping the soil microbial communities of which they are a part, as previously reported [72]. On the other hand, the bacterial use of amino acids, carboxylic acids and polymers is positively related to β-glucosidase and acid phosphatase. Moreover, the use of amines and phenolic compounds by bacteria is positively related to glucosidase and alkaline phosphatase, respectively. These correlations are not significant for fungal communities, suggesting a dominant role of the bacterial communities in driving the main enzymatic reactions in the experimental conditions. These results are apparently in contrast with some previous findings. For example, β-glucosidase activity in soil was reported to be mainly promoted by fungi rather than bacteria [73]. However, this is not totally surprising, as different soil microbial groups in the community are responsible for specific functions. For instance, soil fungi mostly participate at the beginning of litter decomposition, while bacteria play the primary roles at later stages [74].

### 4.4. Bacterial and Fungal Community Structure

The analysis of the microbial community structure showed that each original soil hosted distinct dominant bacterial and fungal communities which developed differently when reinoculated with a different soil type (Figures 4 and S2). This result is apparently not in accordance with the findings of Delmont et al., who reported that distinct communities evolved similarly when colonizing the same sterilized soil [20]. However, in that work the original soils had a similar community composition (richness) and thus developed similarly when colonizing the same habitat. Moreover, the authors incubated the soils for as long as 24 weeks. However, they also observed that the 'new' communities displayed some previously undetected species when colonizing the same sterilized soil, thus providing additional information on the importance of rare microbial species in soil. In fact, it has been widely reported that soil physicochemical properties are the main factors controlling the composition and diversity of soil bacterial communities and that the dominant microbial groups play an active role in the soil functions [75–78]. Genetic fingerprinting showed that bacterial communities evolve much faster than fungal communities during the colonization of soil, likely due to their faster reproduction cycles. A significant fraction of soil microbial biomass belongs to a relatively small number of predominant species, while the majority of microbial species is present in low numbers or in a reversible state of dormancy or reduced metabolic activity [79]. It has been reported that dormant microorganisms or rare species can be trigged into activity in the presence of appropriate substrates and growth conditions [80,81], and, overall, our results support the 'dormant seed' hypothesis for soil microbial communities [82]. The fact that the sterile cross-reinoculated soils generally displayed significantly higher enzymatic activities than non-reinoculated soils and self-reinoculated soils suggests that the exogenous soil microbial communities exploited their potential functionality given the wide availability of new organic substrates. The soil reinoculation approach confirmed the loose relationship between soil microbial diversity and enzymatic activity and that soil properties and substrate availability are the main factors inducing the release of microbial enzymes in soil. It also confirmed the key role of rare microbial species in promoting soil enzymatic activity [83]. Monitoring of the microbial communities over time confirmed that bacteria are more sensitive than fungi to soil properties [61,75], with soil pH playing a major role in shaping the microbial communities [48].

### 4.5. The Effect of Inocula and Substrates

Previous studies reported that soil properties rather than microbial inocula were more important in determining the microbial biomass, bacterial composition and enzymatic activity of sterilized and inoculated soils after 8 months of incubation [71]. Accordingly,

our results showed that the major effect on enzyme activity, microbial catabolic profile and community structure seemed to be due to the original soils rather than to the inocula. In fact, the self- and cross-inoculations provided to each soil did not eliminate the effects of the native soil features, even though there were some differences among them. Moreover, the different inocula displayed increasing effects in shaping Rom, Vic and Val soils, respectively. This was likely due to the different physicochemical characteristics of the three soils, according to Meola et al. [84]. However, stronger community shifts and greater variations among the replicates were not observed after reinoculation of nutrient-rich soils, such as Vallombrosa, in contrast with some previous findings [22]. On the other hand, in a similar study, Kapagianni et al. [23] showed that the availability of C and N after sterilization of different type of soils may vary not only depending on the absolute amounts of organic C and N but also on the quality of the organic matter contained in the different soils.

We could speculate that at the beginning of incubation, in sterilized soils there was no limitation of nutrient availability and a lack of competition with more adapted microbes. Thus, the first colonization step might have been performed by fast growing taxa (r strategists). In fact, our results showed that β-glucosidase was positively correlated with the overall bacterial catabolic activity (AWCD), but not with fungal AWCD (Figure 6). This is not surprising, as fungi have always been considered the major decomposers of recalcitrant organic matter in soil environments, whereas bacteria have been reported to play a major role in the degradation of simple substrates. However, based on the ECO and FF microplate results of this work, both bacterial and fungal catabolic activity in all the sterilized soils increased after 7 and 30 days, thus indicating that they were both metabolically active since a very few days after the sterilization. This result is in accordance with previous studies revealing that the fungal contribution to the decomposition of easily degradable substrates may be high, especially in acidic soils, and at high substrate loading rates [85]. However, here the fungal catabolic rate 's' was negatively correlated with alkaline phosphatase, suggesting that such activities are much more closely related to fungal metabolic activity. Thus, the presence of fungi with the ability to rapidly decompose easily degradable organic compounds must exert a selection pressure on bacteria to compete for these nutrients. Thus, the faster metabolic potential of bacterial communities likely succeeded in promoting alkaline phosphatase and β-glucosidase activity. The combined results highlighted that the interaction between bacteria and fungi is essential to drive metabolic processes in complex environments, such as soil.

## 5. Conclusions

The combination of self- and cross-inoculation of different heat-sterilized soils produced rapid and dynamic changes in enzymatic activity as well as in microbial structure and catabolic activity. Original soils had the major influence in shaping soil functional diversity, while the effect of reinoculation of sterilized soils was more related to incubation period and type of soil. For example, the enzymatic and catabolic activity of pH neutral soils (Rom) increased and decreased after reinoculation with alkaline and acidic soils, respectively. In contrast, acidic (Val) and alkaline (Vic) soils did not show such pH-related responses. In general, Val soil displayed the greatest functional changes after sterilization and reinoculation compared to original untreated soil.

Overall, alkaline phosphatase activity was more closely correlated to fungal catabolic potential, whereas β-glucosidase and acid phosphatase were mainly correlated with bacterial metabolism, thus suggesting that they might have been involved in the breakdown of the organic materials released after sterilization. In contrast, at the early stage of soil recolonization, both protease and urease activities were poorly related to microbial catabolic potential and not significantly affected by the different treatments.

The assembly of early-stage microbial communities in reinoculated soils provided a different structure compared to the pristine soils, showing faster and greater changes in the bacterial community structure than in fungal communities, especially in acidic soils. Interestingly, these 'newly assembled' microbial communities revealed the occurrence of

taxa which were not detected in the original soils, suggesting the key role of rare species during the first colonization phase of a new environment. This result confirmed that rare microbial taxa rather than the dominant taxa may be the major drivers of soil functionality, confirming that rare taxa have a crucial role in biological processes and the sustainable provision of ecosystem functions in the future [20,86].

In conclusion, even though they have inherent limitations, reinoculation experiments have the potential to explore the main rules of microbial adaptation during the early phases of soil recolonization after a severe environmental disturbance as well as the potential to facilitate the discovery of novel rare microorganisms.

**Supplementary Materials:** The following are available online at https://www.mdpi.com/article/10.3390/agriculture12020268/s1, Figure S1: dsDNA content (ex-pressed as mg/kg) of Vallombrosa, Vicarello and Romola control and inoculated soils, Figure S2: DGGE profiles of 16S (A) and 18S (B) rRNA genes PCR products obtained from Vallombrosa, Vicarello and Romola control and inoculated soils, Table S1: Correlations.

**Author Contributions:** Conceptualization, G.R., A.G. and S.M.; methodology, G.R., A.G. and S.M.; formal analysis, G.R., A.G., S.M., L.G., R.P. and B.P.; data curation, S.M.; writing—original draft preparation, S.M., G.R., A.G. and R.P.; writing—review and editing, S.M., G.R. and A.G.; supervision, P.N. All authors have read and agreed to the published version of the manuscript.

**Funding:** This research received no external funding.

**Institutional Review Board Statement:** Not applicable.

**Informed Consent Statement:** Not applicable.

**Data Availability Statement:** The data presented in this study are available on request from the corresponding author.

**Conflicts of Interest:** The authors declare no conflict of interest.

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
