# Peer review of "Short-Term Resilience of Soil Microbial Communities and Functions Following Severe Environmental Changes"

_agriculture, doi:10.3390/agriculture12020268_

Round 1

Reviewer 1 Report

Overall, the article is well organized and its presentation is good. This paper answers two main questions through a systematic experimental analysis: i) do soil microbial communities recover to their original structure and functionality after drastic disturbance? ii) do same microbial communities colonizing different soils express similar 85 biochemical functions? The paper helps to deepen the study of microbial resilience and microbial metabolic activity under extreme environmental changes and it is then of interest for publication in Agriculture.

The value of this paper should be more clearly stressed. Nevertheless results obtained in this paper are rather good and then the paper presents a good level of impact and reference value.

  1. The abstract should be improved, to show the significance of this paper. The abbreviation “ATP” and “CLPP” in the abstract should illustrate.
  2. In line 2, the title of this paper only reflects the research methodology and content, but not its core research significance. If feasible, the title of the article should ideally reflect the research significance of the article, such as “Short-term resilience of soil microbial communities and functions following self and cross re-inoculation of heat-sterilized soils under extreme environmental change”.
  3. In line 21, “environmental change” has a broad meaning, whether it is mild environmental change or extreme environmental change, and there may be differences in the study of soil microorganisms under different types of environmental change. From this paper, it is a study of microbial resilience behavior under severe disturbance, so it is the extreme harsh environmental change, or environmental change under severe disturbance, that needs to be reflected here. The objectives of the study in this paper will be more focused.
  4. In line 26, why was the soil incubation chosen at 1,7,30 days. The Materials and Methods section does not mention the reason for the choice and needs to be clarified. The subject is short-term resilience, why the changes in microbial communities and activities at shorter incubation times, e.g. 1, 5, 10, 20, 30 days, have not been studied?
  5. In line 26, the abbreviation “CLPP” should appear in the preceding text.
  6. In line 30, does “recipient soil” refer to “sterilized soil” or “recolonized soil”? Need to be consistent with the context.
  7. In line 34, since the newly introduced microbial communities was not detected in the original soil, it is better to write the name of the newly discovered microbial communities here to reflect its importance.

Author Response

Reviewer#1

Overall, the article is well organized and its presentation is good. This paper answers two main questions through a systematic experimental analysis: i) do soil microbial communities recover to their original structure and functionality after drastic disturbance? ii) do same microbial communities colonizing different soils express similar biochemical functions? The paper helps to deepen the study of microbial resilience and microbial metabolic activity under extreme environmental changes and it is then of interest for publication in Agriculture.

The value of this paper should be more clearly stressed. Nevertheless results obtained in this paper are rather good and then the paper presents a good level of impact and reference value.

  1. The abstract should be improved, to show the significance of this paper. The abbreviation “ATP” and “CLPP” in the abstract should illustrate.

Thank you for your valuable comment. Accordingly, we tried to improve the abstract by better highlighting the significance of the study (lines 32-34). We also illustrated the significance of ATP and CLPP abbreviations, as suggested (lines 22-23)

  1. In line 2, the title of this paper only reflects the research methodology and content, but not its core research significance. If feasible, the title of the article should ideally reflect the research significance of the article, such as “Short-term resilience of soil microbial communities and functions following self and cross re-inoculation of heat-sterilized soils under extreme environmental change”.

     We appreciate the comment and we agree with the opinion of the reviewer. However, the suggested change appeared to be somehow redundant. Therefore, we modified the title by replacing a methodological information with another one more focused on the research significance, according to your suggestion: ”Short-term resilience of soil microbial communities and functions following severe environmental changes”

  1. In line 21, “environmental change” has a broad meaning, whether it is mild environmental change or extreme environmental change, and there may be differences in the study of soil microorganisms under different types of environmental change. From this paper, it is a study of microbial resilience behavior under severe disturbance, so it is the extreme harsh environmental change, or environmental change under severe disturbance, that needs to be reflected here. The objectives of the study in this paper will be more focused.

     We agree with the comment and modified the text accordingly, specifying “extreme environmental changes”. As mentioned before, we also modified the text to better focusing the main research objectives, as described along the introduction section of the paper (lines 88-90).

  1. In line 26, why was the soil incubation chosen at 1,7,30 days. The Materials and Methods section does not mention the reason for the choice and needs to be clarified. The subject is short-term resilience, why the changes in microbial communities and activities at shorter incubation times, e.g. 1, 5, 10, 20, 30 days, have not been studied?

     Thank you for this comment. Accordingly, we specified the rationale behind the choice of the incubation times (lines 174-176). First, we based our choice considering the experimental setup previously adopted by other authors for similar purposes (es. Dalmont et al., 2014; Francioli et al., 2016; Kapagianni et al., 2019), who showed relevant results occurring between 25 and 90 days of incubation. Thus, in order to spread some light to the very first functional changes occurring in soils after the disturbance, we selected a 30d-incubation period because, at the best of our knowledge, it has never been adopted in similar experiments. Second, we did not include other shorter incubation times (es. 1,5, 10, 20) because we were not sure being able to get some relevant result by incubating samples for less than 30 days. So, also considering the huge amount of work requested for each incubation time, we decided to focus on samples incubated for 1, 7 and 30 days. Of course, in the future it will be interesting to focus the research on just few specific soil functions/activities or microbial community structure to be assessed over more frequent incubation times, as suggested.

  1. In line 26, the abbreviation “CLPP” should appear in the preceding text.

We agree with this comment. As mentioned above, we modified the text accordingly.

  1. In line 30, does “recipient soil” refer to “sterilized soil” or “recolonized soil”? Need to be consistent with the context.

  Thank you for highlighting this point. “Recipient” soil refers to the original soils that have been sterilized. In order to avoid any possible confusion, we’ve added some clarification in the M&M section.

  1. In line 34, since the newly introduced microbial communities was not detected in the original soil, it is better to write the name of the newly discovered microbial communities here to reflect its importance.

We understand the comment, but we cannot specify the names of the different microbial taxa because the DGGE technique we used here just provides the overall structure of the microbial community, highlighting the dominant species/strains (see also Figure S1). But we agree with the comment and this is something we’re planning to do in the next future.

Reviewer 2 Report

Revision Agriculture / Manuscript ID: 1561368

The Authors presented very comprehensive and time consuming research that takes into account a wide variety of factors.

Comments and suggestions for Authors:

Introduction:

This manuscript is content of broad international interest and the aim of the study is clearly stated. This manuscript possess scientific and practical values.

Materials and methods:

The manuscript contains quite carefully described methodology.

Results:

Tables and figures:

Figure 1 : the legend and the letters indicating the significant differences among means are too small, also some scales need to be unify, to compare the results (e.g. Figure 1 A Vallobrosa and Vicarello should have the same scale and Vallombrosa has value 300 on a scale so it could be the maximum value on Romola scale).

Table 3 and Table 4: I suggest to present this results on a figures. It is very hard to compare these results and ‘draw’ some conclusions (and follow the Authors conclusions).

Discussion:

The discussion interpretate the results and shows how these results relates to the literature review. Authors should indicate the numbers of Tables and Figures, where ever required (where ever the results are mentioned). All (own) results needs to be mentioned and discussed.

References:

The ‘References’ part needs to be prepared in accordance with the principles of editing.

Author Response

The Authors presented very comprehensive and time consuming research that takes into account a wide variety of factors.

Comments and suggestions for Authors:

Introduction:

This manuscript is content of broad international interest and the aim of the study is clearly stated. This manuscript possess scientific and practical values.

Thank you for your kind comment.

Materials and methods:

The manuscript contains quite carefully described methodology.

Results:

Tables and figures:

Figure 1: the legend and the letters indicating the significant differences among means are too small, also some scales need to be unify, to compare the results (e.g. Figure 1 A Vallobrosa and Vicarello should have the same scale and Vallombrosa has value 300 on a scale so it could be the maximum value on Romola scale).

Thank you very much for this comment. Accordingly, we modified Fig.1 in order to make both legends and letters more readable. Moreover, scales in Fig.1A and Fig1B were unified, when appropriate. In fact, in some cases we preferred to keep a different scale to better visualize the results which would have been hidden in a different scale (e.g. Fig.1A Romola, Fig.1B Vallombrosa, Fig.1C Vicarello).

We also replaced Fig.5, as the names of the variables on the left were partially out of the figure. Thus, we changed the names with abbreviations (specifying the meaning in the caption).

Table 3 and Table 4: I suggest to present this results on a figures. It is very hard to compare these results and ‘draw’ some conclusions (and follow the Authors conclusions).

In principle, we agree with the comment. However, tables 3 and 4 include a lot of data with several variables which make it difficult to represent the entire results in one or few clear figures. After several attempts, we obtained 12 figures (one for each CLPP substrates for bacteria and fungi), but the displayed data and the information was too small and poorly readable. Therefore, we’d prefer to leave such results on tables.

Discussion:

The discussion interpretate the results and shows how these results relates to the literature review. Authors should indicate the numbers of Tables and Figures, where ever required (where ever the results are mentioned). All (own) results needs to be mentioned and discussed.

As suggested, we carefully checked all the subsections of the discussion, to be sure that all the results were mentioned and discussed, with the proper indication to the numbers of Tables/Figures.

In section 4.1 “Soil respiration, N ammonification, ATP and microbial biomass” all the main results obtained in this work were mentioned and discussed, referring to Figure 1 (but also Table 2 and Figure 3), except for the data of microbial biomass (dsDNA). In fact, although we used dsDNA data for correlation and multivariate analysis (PCA) and final discussion, we did not include it into the results and discussion sections. We really apologize for this. So, we added the missing information in the Result section (lines 292-298) and in the Discussion (lines 673-675), as well as an additional figure included in the supplementary materials (Figure S2).

In section 4.2 “Soil enzymatic activities” we discussed data included in Table 2. In addition, we specified the Tables and Figures referred to microbial biomass and CLPP mentioned in the second part of the section (lines 715-716).

In section 4.3 “Soil community level physiological profile” we discussed about CLPP result. Thus, we added an indication to Figure 2 as well as to Tables 3 and 4 (line 774) and Figure 5 (line 791) which have not been mentioned in the previous version.

In section 4.4 “Bacterial and fungal community structure” we added a reference to Figure S1 (line 811) for the discussion of microbial community structure and in section 4.5 “The effect of inocula and substrates” we discussed about the results of the multivariate analysis, adding a specific mention to Figure 6.

References:

The ‘References’ part needs to be prepared in accordance with the principles of editing.

We modified the entire reference section, according to the journal’s indications.